# Acceptance rate and risk perception towards the COVID-19 vaccine in Botswana

**Lebapotswe B. Tlale**[1]*, **Lesego Gabaitiri**[2], **Lorato K. Totolo**[1], **Gomolemo Smith**[1], **Orapeleng Puswane-Katse**[1], **Eunice Ramonna**[1], **Basego Mothowaeng**[1], **John Tlhakanelo**[2], **Tiny Masupe**[2], **Goabaone Rankgoane-Pono**[1], **John Irige**[3], **Faith Mafa**[1], **Samuel Kolane**[1]

**1** Ministry of Health and Wellness, Gaborone, Botswana, **2** Botswana International University of Science and Technology, Palapye, Botswana, **3** FHI 360, Gaborone, Botswana

* cypro330@yahoo.com, Lebapotswe.tlale@gmail.com

**Data Availability Statement:** All relevant data are within the paper and its Supporting Information files.

## Abstract

### Background

The COVID-19 disease burden continues to be high worldwide and vaccines continue to be developed to help combat the pandemic. Acceptance and risk perception for COVID-19 vaccines is unknown in Botswana despite the government's decision to roll out the vaccine nationally.

### Objectives

This study aims to assess the acceptance rate and risk perception of COVID-19 vaccines amongst the general population in Botswana.

### Methods

We interviewed 5300 adults in Botswana from 1–28 February 2021 using self-administered questionnaires. The main outcomes of the study were vaccine acceptance and hesitancy rates. Demographic, experiential and socio-cultural factors were explored for their association with outcome variables.

### Results

Two-thirds of the participants were females (3199), with those aged 24–54 making the highest proportion (61%). The acceptance rate of COVID-19 vaccine was 73.4% (95% CI: 72.2%-74.6%) with vaccine hesitancy at 31.3% (95% CI: 30.0%-32.6%). When the dependent variable was vaccine acceptance, males had higher odds of accepting the vaccine compared to females (OR = 1.2, 95% CI: 1.0, 1.4). Individuals aged 55–64 had high odds of accepting the vaccine compared to those aged 65 and above (OR = 1.2, 95% CI: 0.6, 2.5). The odds of accepting the vaccine for someone with primary school education were about 2.5 times that of an individual with post graduate level of education. Finally, individuals with comorbidities had higher odds (OR = 1.2, 95% CI: 1.0, 1.5) of accepting the vaccine compared to those without any underlying conditions.

**Funding:** FHI Botswana paid the article publication fee. No additional external funding was received for this study.

**Competing interests:** The authors have declared that no competing interests exist.

## Conclusion

This study demonstrated a high acceptance rate for the COVID-19 vaccine and a low risk perception in Botswana. In order to achieve a high vaccine coverage and ensure a successful vaccination process, there is need to target populations with high vaccine hesitancy rates. A qualitative study to assess the factors associated with vaccine acceptance and hesitancy is recommended to provide an in-depth analysis of the findings.

## Background

According to the World Health Organisation (WHO) weekly epi update of the 14th February 2021, there were a total of 108 246 992 COVID-19 cases and 2 386 717 COVID-19 deaths [1]. Of these, the African region reported a total of 2 723 431 COVID-19 cases and 68 294 deaths, with Botswana reporting 24 926 total cases and 202 total deaths [1]. Vaccines are considered one of the most awaited interventions for combating COVID-19 and hundreds of global institutions are working at an unprecedented speed to develop COVID-19 vaccines [2–9]. Several vaccines have been developed and some are still undergoing clinical trials while very few countries have started vaccine rollout [10]. One of the challenges towards COVID-19 vaccination is the uncertainty of vaccine acceptance among the public. In general, factors that influence vaccine acceptance include the public's demand for vaccine, their perception towards the disease and attitudes towards the vaccine [11].

Acceptance and perception studies are important as they provide critical information which can be used by health education programs to increase the uptake of vaccines and target certain populations. Few studies have been conducted assessing the acceptance, perceptions and attitudes towards COVID-19 vaccines as well as factors influencing uptake which may vary from country to country. A global survey of potential acceptance of a COVID-19 vaccine has shown that differences in acceptance rates ranged from almost 90% (in China) to less than 55% (in Russia) [10]. In an Australian study [12], eighty per cent (80%) respondents generally held positive views towards COVID-19 vaccination while 65% of participants in Saudi Arabia [2], showed interest to accept the COVID-19 vaccine if available. The preliminary results of a Chilean case study on COVID-19 vaccine perception in the country showed that 87% were willing to vaccinate, a relatively high proportion and slightly lower than the rate found by García and Cerda [13] which was 90.6% for Chile. Gender and age disparities were found to have a relation with vaccine acceptance in Australia, where females (83%) were found to be more likely to depict an optimistic outlook to receiving vaccinations than males (78%), as well as those aged 70 years and above (91%) compared to 76% of 18–29-year-olds [12]. Furthermore, willingness to accept future COVID-19 vaccines in Saudi Arabia was relatively high among older age groups, married participants, participants with a postgraduate degree or higher education level (68.8%), non-Saudi (69.1%), employed in government sector (68.9%) [2]. There is subcultural diversity as African Americans and Hispanics demonstrated higher vaccine hesitancy than other cultural or ethnic groups in a US study [14].

Religion, female gender, residing in deprived neighbourhoods were some factors found to be correlated with COVID-19 vaccine hesitancy in Australia [15], while reliance on social media and refraining from news [16] were also associated with vaccine hesitancy in the UK.

These studies demonstrate significant variations in COVID-19 vaccine acceptance across different countries and the roles of several socio-demographic determinants of health in vaccine acceptance and risk perception. Such information is critical for context and country

specific implementation of COVID-19 vaccination programs, with governments being encouraged to understand communities' concerns and identify strategies that will support engagement to support effective launching of new vaccines [12].

Circumstantial conditions such as the pandemic context, specifically disease prevalence in a particular population can also impact vaccination intention [13]. Perceived severity of COVID-19 and perceived vaccine safety were the two strongest determinants of vaccine acceptance in a Finland study [17]. In Turkey and the UK, acceptance rates of vaccines were found to be higher among study participants who believed in the natural origin of COVID-19 in contrast to those who believed that the disease was generated by humans [18]. Moreover, credible sources of information about vaccines such as government were reported to instill high levels of trust amongst the public [10].

In the African setting, the level of vaccine acceptance (53.6%) and risk perception of (46.7%) were relatively average in Western Uganda [19]. Males, those with a tertiary education, students and non-salary earners were likely to accept the vaccine [19].

The Botswana government has adopted a multi-pronged strategy as part of the response to COVID-19. One of the pillars of the COVID-19 epidemic control measures includes vaccinating 276, 078 (16.5%) of the targeted population in the 1st phase of the vaccination campaign. Botswana is a signatory to the WHO/World bank vaccine initiative and expects to receive its first COVID-19 vaccines in March 2021. Acceptance and risk perception for COVID-19 vaccines is unknown in Botswana. There have been several conspiracy theories around COVID-19 vaccines which are mostly linked to religious and cultural beliefs which may influence the uptake of the vaccine [20]. The widely used social media platforms have also reported on the negative aspects of the vaccine and its side effects which may also influence the population's attitudes and perceptions resulting in low acceptance [21]. We therefore assessed the acceptance and risk perceptions of COVID-19 vaccines in Botswana in order to inform the planned population roll out of the vaccines. Demographic, experiential and socio-cultural factors were also explored for their association with the outcome measures.

## Materials and methods

### Study design

This study used a cross sectional survey design conducted across the nine (9) COVID-19 zones in Botswana from 1–28 February 2021 using enumerator administered questionnaires designed from literature review and using a risk perception model which integrates three core dimensions; cognitive factors (knowledge), experiential factors (emotion) and socio-cultural factors (norms, values) [12, 22, 23]. The questionnaires were self-administered without any interference from the interviewer.

To ascertain quality, the questionnaire was pretested before the final draft was made. The development of the questions was guided by the WHO Technical Advisory Group on Behavioural Insights and Sciences for Health paper entitled "Behavioural Considerations for Acceptance and Uptake of COVID-19 Vaccines". The draft contained about 7 themes and some were dropped because some of the questions were similar or a repetition of questions in other themes. The final version contained Demographic information, Cognitive Factors, Experiential Factors, and Socio-cultural factors. The final version was translated into the native official language and back into English language.

### Study setting

Botswana is a landlocked country situated in Southern Africa and shares borders with South Africa, Namibia, Zimbabwe and Zambia. The dynamics of population movement across

borders with neighbouring countries increases the risk of transmission of communicable diseases including COVID-19. Botswana has an estimated population of 2.374,697, (1,171,629 Males and 1,203,068 Females) as per the 2011 population census projection for 2020 [24]. The population is spread over vast land of 581,730km$^2$. The estimated population density in Botswana is 4 per Km. Sixty nine percent (69.4%) of the total population lives in urban settings.

Botswana's economy is largely dependent on mineral revenues and belongs to Southern Africa Custom Union. The Gross National Income (GNI) per capita is approximately $6,000, thus by classification, Botswana belongs to the upper middle-income countries. Botswana recorded its first COVID-19 case in March 2020. After recording the first case the country was demarcated into nine (9) sections called COVID-19 zones in order to restrict movement of non-essential travel between these zones. In order to travel between these zones, a valid permit, which should be applied for online, is necessary.

## Participants

The study population consisted of 5300 adults aged 18 years and above since most candidate vaccines were investigated and tried in this group worldwide. In addition, this was in line with the Botswana government policy for COVID-19 vaccine eligibility. Study participants were selected from localities in each of the nine (9) Botswana COVID-19 zones using stratified sampling. Eligibility for participation in the study included those aged 18 years and above including pregnant women, and those with (and without) comorbidities and able to give informed consent. Following consent, participants were enrolled into the study by administering the questionnaires in the selected localities. This happened at government departments, shops, markets and private companies. Exclusion criteria included those who are under 18 years of age and those that did not consent to participate in the study.

## Study size

The survey leveraged on the already existing resources. The nine (9) COVID-19 zones were used as strata (Table 1). A stratified sampling was employed for this survey. Individuals/study participants were selected within the localities. The localities were selected from the COVID-19 zones.

To compute the sample size for the survey a margin of error of 0.05 and a confidence level of 0.95 were used. We further assumed an acceptance rate of 53.6% which is similar to the one

**Table 1. The 9 COVID-19 zones, sample size and allocation.**

| Zone | Population | | Proportional Sample | |
| --- | --- | --- | --- | --- |
| | Localities | > = 18 years | Locality | Individuals |
| Greater Gaborone | 167 | 590,689 | 3 | 2712 |
| Greater Palapye | 78 | 188,428 | 2 | 865 |
| Maun | 52 | 84,449 | 1 | 388 |
| Greater Francistown | 80 | 192,483 | 2 | 884 |
| Chobe | 18 | 23,789 | 1 | 109 |
| Ghanzi | 17 | 22,474 | 1 | 103 |
| Kgalagadi | 47 | 30,678 | 1 | 141 |
| Selebi Phikwe | 21 | 74,678 | 1 | 343 |
| Boteti | 20 | 43,810 | 1 | 201 |
| Total | 500 | 1,238,768 | 13 | **5745** |

estimated by Echoru I et al (2020) in Uganda [19] resulting in a sample of size 383. To cover the 9 COVID-19 zones, we estimated a sample size of 383 multiplied by nine (9) zones giving a total of 3447. To accommodate the design effect, since our sampling strategy is stratified sampling scheme, we assumed a design effect of 1.5 which then gave us a sample size of 5171. Finally, we assumed 90% response rate which resulted in a total sample of **5745** for the survey.

## Variables

The exposure variables for this study included age, gender, level of education, Occupation, religious background, Media, marital status and place of residence. The primary outcome measure includes acceptance rate and risk perception towards the COVID-19 vaccine.

## Data sources

The cross-sectional survey used questionnaires to interview participants from selected areas across the country. The questionnaires were written in both English and Setswana and data obtained from the questionnaire were analysed using the STATA statistical software version 15. The questionnaires were validated before being administered to all the participants. The participants answered the same questionnaire across zones to obtain reliable information. Data protection was ensured via file password protection and limiting access of information to the project team members only.

## Statistical methods

Categorical variables were analysed using summary statistics such as frequencies and percentages. For continuous variables such as age, median (inter-quartile range) or mean (and standard deviation) were presented depending on whether the variable under consideration is skewed or not, respectively. Acceptance was measured by the question asking respondents if they are willing to receive the COVID-19 vaccine when it is rolled out nationally. The acceptance rate of COVID-19 vaccine was the number that "accept" divided by the sample size but excluding those with missing responses. The independent variables were demographic variables such as age, gender, marital status, educational level, and other important variables such presence and absence of comorbidities, COVID-19 zone level (high/red zone, medium, low), socio-cultural and experimental factors. Risk perception was measured by the questions asking respondents if they considered the COVID-19 vaccine to be safe.

Bivariate analysis using the chi-square test was conducted to determine factors associated with both acceptance and risk perception for the COVID-19 vaccine. Multiple logistic regression was conducted to adjust for any confounding factors. Odds ratios, as measures of effect, and the corresponding p-values and 95% confidence intervals are presented. A p-value less than 0.05 was used to determine significance.

## Ethical considerations

This study was conducted according to Botswana, and International Standards of Good Clinical Practice, applicable government regulations and Institutional research policies and procedures. The protocol and any amendments made were submitted to the Botswana, Ministry of Health and Wellness Institutional Review Board (IRB), were it was approved. The protocol number awarded for this study is HPDME 13/18/1.

All subjects were asked to provide written informed consent before responding to questions and they were free to withdraw at any time during the study.

## Results

### a. Demographic details of participants

The total number of participants interviewed was 5300. Two-thirds were females, with the age group 24–54 years accounting for the highest proportion. Majority of the participants had attended senior secondary school, were employed, had no comorbidities and were Christians (Table 2).

**Table 2. Socio-demographic characteristics of participants (n = 5300).**

| Gender | Total | Percentage |
|---|---|---|
| Male | 1904 | 35.9% |
| Female | 3347 | 63.2% |
| Missing | 49 | 0.9% |
| **Age**[*] | | |
| 18–24 | 774 | 1.64% |
| 25–54 | 3366 | 63.5% |
| 55–64 | 281 | 5.3% |
| 65 and above | 115 | 2.2% |
| Missing | 764 | 14.4% |
| **Marital Status** | | |
| Single | 4014 | 75.7% |
| Married | 1046 | 19.7% |
| Divorced | 99 | 1.9% |
| Widowed | 83 | 1.6% |
| Missing | 58 | 1.1% |
| **Religion** | | |
| Christian | 4777 | 90.1% |
| Hindu | 33 | 0.6% |
| Islam | 57 | 1.1% |
| Buddhism | 11 | 0.2% |
| Other | 271 | 5.3% |
| Missing | 143 | 2.7% |
| **Residence** | | |
| Rural | 2334 | 44.0% |
| Semi-Urban | 1482 | 28.0% |
| Urban | 1367 | 25.8% |
| Missing | 117 | 2.2% |
| **Education level** | | |
| Primary | 471 | 8.9% |
| Junior Secondary | 1336 | 25.2% |
| Senior Secondary | 1575 | 29.7% |
| Under Graduate | 701 | 13.2% |
| Post Graduate | 990 | 18.7% |
| Missing | 227 | 4.3% |
| **Willingness to wear mask** | | |
| Yes | 4738 | 89.4% |
| No | 227 | 5.2% |
| Missing | 285 | 5.4% |
| **Employment status** | | |

(*Continued*)

**Table 2.** (Continued)

| Gender | Total | Percentage |
|---|---:|---:|
| Employed | 2893 | 54.6% |
| Unemployed | 1714 | 32.3% |
| Student | 363 | 6.8% |
| Others | 282 | 5.3% |
| Missing | 48 | 0.9% |
| **Medical history** | | |
| Comorbidities | 1332 | 25.1% |
| Non-comorbidities | 3770 | 71.1% |
| Missing | 198 | 3.7% |
| **Districts** | | |
| Boteti | 269 | 5.1% |
| Chobe | 180 | 3.4% |
| Gantsi | 137 | 2.6% |
| Greater Francistown | 1154 | 21.8% |
| Greater Gaborone | 1501 | 28.3% |
| Greater Palapye | 873 | 16.5% |
| Greater Phikwe | 466 | 8.8% |
| Kgalagadi | 201 | 3.8% |
| Maun | 519 | 9.8% |

[*]age categorised in reference to Lazarus J V et al (2021) [10].

## b) Acceptance rate for COVID-19 vaccine in Botswana

Three thousand, six hundred and eighty nine (3689) out of five thousand and twenty seven (5027). Participants were willing to take the vaccine resulting in the acceptance rate of the COVID-19 vaccine in Botswana of 73.4% (CI: 72.2%, 74.6%).

## c. Factors associated with acceptance of the COVID-19 vaccine in Botswana

Table 3 shows the results of adjusted logistic regression with selected variables, similar to those in Table 1 except district. We left district out the model because it has many categories which will lead to too many parameters to be estimated in the model. Two models were fitted with different outcomes variables. The first model focused on factors associated with the acceptance rate of the COVID-19 vaccine while the second model focused on factors associated safety perception.

When the dependent variable is vaccine acceptance the following variables were significantly associated the outcome; sex, age, education level, willingness to wear mask, employment status and medical history. Generally, males have higher odds of accepting the vaccine compared to females (OR = 1.2, 95% CI: 1.0, 1.4). Individuals aged 55–64 have high odds of accepting the vaccine compared to those aged 65 and above (OR = 1.2, 95% CI: 0.6, 2.5). However, those aged below 55 years have lower odds of accepting the vaccine compared to those aged 65 and above. The odds of accepting the vaccine for someone with primary school education are about 2.5 times that of an individual with post graduate level of education. Finally, individuals with comorbidities have higher odds (OR = 1.2, 95% CI: 1.0, 1.5) of accepting the vaccine compared to those without any underlying conditions.

We also investigated factors associated with safety perception. Generally, the significant factors associated with safety perception are similar to those associated with vaccine

**Table 3. Factors associated with COVID-19 vaccine acceptance rate and safety perception among participants in Botswana.** (n = 5300).

| | ACCEPTANCE | | SAFETY PERCEPTION | |
| --- | --- | --- | --- | --- |
| | Adjusted Model results | | Adjusted Model results | |
| | OR (95% CI) | P-value | OR (95% CI) | P-value |
| **Sex** | | | | |
| Male | 1.2 (1.0, 1.4) | 0.032** | 1.0 (0.8, 1.1) | 0.663 |
| Female | 1 | | 1 | |
| **Age*** | | 0.037** | | 0.031** |
| 18–24 | 0.7 (0.4, 1.3) | | 0.7 (0.3, 1.3) | |
| 25–54 | 0.9 (0.5, 1.7) | | 0.8 (0.4, 1.5) | |
| 55–64 | 1.2 (0.6, 2.5) | | 1.2 (0.6, 2.4) | |
| 65 and above | 1 | | 1 | |
| **Marital Status** | | 0.425 | | 0.496 |
| Single | 1.5 (0.8, 3.0) | | 1.6 (0.8, 3.1) | |
| Married | 1.3 (0.7, 2.6) | | 1.6 (0.8, 3.1) | |
| Divorced | 1.4 (0.6, 3.5) | | 1.3 (0.6, 3.1) | |
| Widowed | 1 | | 1 | |
| **Religion** | | | | |
| Christian | 1.1 (0.8, 1.5) | 0.462 | 1.1 (0.8, 1.5) | 0.555 |
| Other | 1 | | 1 | |
| **Residence** | | 0.158 | | 0.079 |
| Rural | 0.9 (0.8, 1.1) | | 0.9 (0.8, 1.1) | |
| Semi-Urban | 0.8 (0.7, 1.0) | | 0.8 (0.7, 1.0) | |
| Urban | 1 | | 1 | |
| **Education level** | | <0.001** | | <0.001** |
| Primary | 2.5 (1.7, 3.7) | | 2.2 (1.6, 3.2) | |
| Junior Secondary | 1.8 (1.4, 2.3) | | 2.0 (1.6, 2.5) | |
| Senior Secondary | 1.3 (1.1, 1.6) | | 1.2 (1.0, 1.5) | |
| Under Graduate | 0.8 (.7, 1.1) | | 0.9 (0.7, 1.1) | |
| Post Graduate | 1 | | 1 | |
| **Willingness to wear mask** | | | | |
| Yes | 3.5 (2.6, 4.7) | <0.001** | 2.9 (2.2, 4.0) | <0.001** |
| No | 1 | | 1 | |
| **Employment status** | | 0.035** | | 0.192 |
| Employed | 1.2 (0.9, 1.7) | | 1.1 (0.8, 1.5) | |
| Unemployed | 1.5 (1.1, 2.1) | | 1.3 (0.9, 1.8) | |
| Student | 1.2 (1.0, 1.5) | | 1.1 (0.7, 1.7) | |
| Others | 1 | | 1 | |
| **Medical history** | | | | |
| Comorbidities | 1.2 (1.0, 1.5) | 0.030** | 1.2 (1.0, 1.4) | 0.094 |
| Non-comorbidities | 1 | | 1 | |

** Statistically significant P <0.05.

acceptance except sex, employment status and medical history which are no not signifi-
cant.). Individuals aged 55–64 believe the vaccine is safe to use compared to those aged 65
and above (OR = 1.2, 95% CI: 0.6, 2.4). Also, individuals willing to wear mask believe the
vaccine is safe.

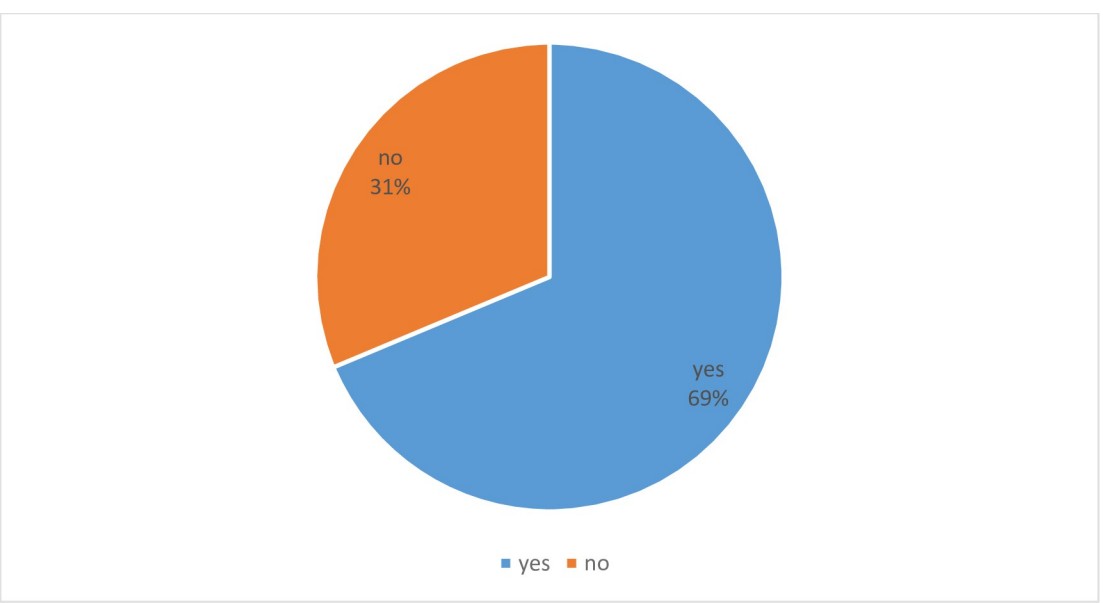

**Fig 1. Vaccine safety.**

### d. Vaccine safety

Fig 1 shows that out of 4784 participants, 1499 (31.3%) (CI: 30.0%, 32.6%) of participants believed that the COVID-19 vaccine was not safe.

### e. Willingness to take the vaccine against religion and culture

Participants whose religious beliefs did not hinder vaccination were more likely to take the vaccine than those whose religious beliefs hinder vaccination. Table 4 shows that about half (49%) of the participants who said their religious and cultural beliefs hinder vaccine uptake were willing to receive the COVID-19 vaccine. Additionally the majority (77.6%) of those who had no hindrances from religious and cultural beliefs, were willing to receive the COVID-19 vaccine. The odds of willing to receive COVID-19 vaccine are lower for those who indicated that religion or cultural beliefs hinder vaccine uptake compared those who did not (OR = 0.3, 95% CI: 0.2, 0.3). About fifty-one percent (51%) of participants who trusted other traditional and religious methods over the vaccine were willing to receive the COVID-19 vaccine. Also, we note that the odds of accepting vaccine are low for those who trust other traditional and religious methods over the vaccine (OR = 0.3, 95% CI: 0.2, 0.3).

**Table 4. Willingness to receive COVID-19 vaccine against cultural and religious believes among participants in Botswana.**

| | | Willingness to receive COVID-19 vaccine | | | | |
| --- | --- | --- | --- | --- | --- | --- |
| | | **Yes** | **No** | **Total** | **P value** | **Odds (95% CI)** |
| Said religion or cultural beliefs hinder vaccine uptake | Yes | 361 (49.2%) | 372 (50.8%) | 724 (100%) | <0.001 | 0.3 (0.2, 0.3) |
| | No | 3326 (77.6%) | 962 (22.4%) | 4228 (100%) | | |
| | Total | 3687 (73.4%) | 1334 (26.6%) | 4952 (100%) | | |
| Trust other traditional and religious methods over the vaccine | Yes | 607 (51.4%) | 574 (48.6%) | 1172 (100%) | <0.001 | 0.3 (0.2, 0.3) |
| | No | 3028 (80.3%) | 741 (19.7%) | 3772 (100%) | | |
| | Total | 3593 (73.4%) | 1301 (26.6%) | 4894 (100%) | | |

**Table 5. The preferred and trusted source of information dissemination among participants in Botswana.**

| Source of information | Preferred source of information dissemination | | Most trusted source of information | | |
| --- | --- | --- | --- | --- | --- |
| | Frequency (%) | Rank | Source of Information | Frequency | Rank |
| Radio | 2664 (50.3) | 1 | Government | 2826 (53.3) | 1 |
| Television | 1412 (26.6) | 2 | WHO | 1892 (35.7) | 2 |
| Social media | 932 (17.6) | 3 | Social media | 1050 (19.8) | 3 |
| Newspaper | 470 (8.9) | 4 | Internet | 435 (8.2) | 4 |
| Internet | 453 (8.5) | 5 | | | |

## f. Preferred and trusted mode of COVID-19 vaccine related communication

Table 5 shows that radio ranked as the number 1 (50.3%) preferred source of information followed by television. The table also shows that the majority (53.3%) of the participants trusted the government most as the source of information for COVID-19, followed by World Health Organization (WHO).

## g. Vaccine acceptance and safety by COVID-19 districts

Table 6 shows vaccine acceptance rate and percentage of participants indicating that the vaccine is safe. Generally, Greater Phikwe zone has both the largest acceptance rate and percentage of participants indicating that the vaccine is safe while Kgalagadi zone has the smallest of both. This therefore means that people in the Kgalagadi zone should be educated about the vaccine safety and be encouraged to up the vaccine when it is available.

## Discussion

Botswana like many other countries around the world is about to introduce free COVID-19 vaccination to its citizens and the acceptance rate (73.4%) from the study is above average and the risk perception (31.3%) is relatively low. In order to achieve herd immunity and prevent hospitalizations as a result of COVID-19, the country needs to vaccinate a high proportion of individuals eligible for vaccination. This requires sufficient vaccine acceptance and low risk perception which can be achieved through social mobilization, health education and health promotion activities targeted towards the eligible groups.

The acceptance rate of 73.4% is similar to that of an online survey conducted in France in May 2020 and from other parts of Europe (Denmark, Germany, Italy, Portugal, the

**Table 6. Vaccine acceptance rate and safety of COVID-19 vaccine by districts in Botswana.**

| Districts/Zones | Acceptance rate | Yes, vaccine is safe |
| --- | --- | --- |
| Boteti | 75.1% | 64.3% |
| Chobe | 65.5% | 65.3% |
| Gantsi | 69.0% | 66.9% |
| Greater Francistown | 75.3% | 70.8% |
| Greater Gaborone | 72.2% | 69.5% |
| Greater Palapye | 73.9% | 68.1% |
| Greater Phikwe | **79.4%** | **76.7%** |
| Kgalagadi | **57.8%** | **55.2%** |
| Maun | 75.2% | 65.6% |

Netherlands, and the UK) [25, 26]. However, the acceptance rate in Botswana is higher than that of Uganda and Russia [10, 19]. The acceptance rate is lower compared to China and countries like Chile [10, 13]. The study team considered these results satisfactory for an upper middle-income country like Botswana which did not participate in any COVID-19 vaccine clinical trials compared to first world countries like the UK and Germany. The difference in acceptance rate among countries could be attributed to many factors such as population dynamics, literacy levels, experience in management of vaccine preventable disease and other factors. The interesting finding in our study is that the acceptance rate is high and risk perception is low. This could likely be due to "intervention fatigue" amongst the community from prolonged compulsory mask wearing laws, nationwide lockdowns and curfews that have been implemented since the beginning of the pandemic, which was earlier and more rigid than in many countries. However, this has not been studied.

Those with comorbidities have been adversely affected by COVID-19 compared to those without comorbidities and from this study we see a high acceptance rate among individuals with comorbidities. This is a population that is at high risk for poor outcomes if infected with COVID-19 [27], therefore it is important that acceptance rate remains high in this group as well as the elderly. Knowing their high risk of severe disease may have contributed to a high acceptance rate in this population. This high acceptance rate is vital for the country's vaccine roll out plan as individuals with comorbidities will be prioritized.

This survey revealed that socio-demographic factors have an impact on the acceptance rate of the COVID-19 vaccine in Botswana. The elderly (55 years and above) had the highest vaccine acceptance rate and this could be associated with the conception that the group pay attention to news through government sources, while the younger groups frequently use social media and internet where there is an array of unverified information and also the knowledge that this population has a higher risk of severe disease. Contrarily, a survey in China showed that middle-aged people (30–49) showed more willingness to take the vaccine than other age groups. According to the authors, factors that affected willingness to be vaccinated included paying close attention to the latest news of the vaccine, among other factors [28].

The majority of the respondents in this study stated that the radio is their most preferred mode of communication over other mass media platforms, social media and internet. They reported government sources as their most trusted sources of information over WHO, social media and internet. Such findings should be encouraging for the implementers that majority of the population is not misinformed by illegitimate sources on social media and the internet. This is also a positive finding as most of the population has access to radio compared to social media and therefore implementers of the COVID-19 vaccine rollout can access the majority of the population through this preferred and accessible platform.

In our study, female participants and Christians were more likely to accept COVID-19 vaccination than the rest of the adult population. Contrary, in China, Japan and Uganda, researchers found that male gender was a significant contributing factor for high vaccine acceptance [11, 19, 29].

Participants who have at least a primary school qualification were more likely to accept the vaccine compared to their counterparts. A Japanese study revealed that the main predictor for vaccine hesitancy is fear of the risk for side-effects [29]. Therefore, it is critical to prioritize educational messages and reassure those who are willing to get vaccinated as well as those who display unwillingness in order to make progress towards herd immunity.

The risk perception of 31.3% is low as compared to that of Uganda, a low-income country in Africa [19]. The reasons why risk perception is low in Botswana have not been studied. However, the Botswana Expanded Program on Immunization (EPI) is old having started in 1979 with high coverages experienced in the past years [30]. Having a robust immunization

program can bring positive outcomes to vaccine uptake and risk perception. Since 1979 when the EPI program was launched, Botswana has introduced a number of vaccines like rotavirus (2012), Measles and Rubella (2016), and this could also be a factor that contributes to a low risk perception.

Ninety six percent of those willing to receive the COVID-19 vaccine indicated that they will continue wearing masks and social distance after vaccination. This is quite high and very important for COVID-19 prevention because the vaccine does not offer absolute protection against COVID-19 [31]. Also half of those who stated that their religion and culture hinder vaccine uptake were willing to receive the covid-19 vaccine. This is very important for health educators as it highlights that it is possible to strengthen public health education, community mobilization and advocacy for behaviour change towards cultural and religious believes and their impact on health outcomes.

## Limitations

This is the first study on acceptance and risk perception towards the COVID-19 vaccine in Botswana. It will provide a foundation for future studies and baseline information for the monitoring of acceptance and risk-perception of COVID-19 vaccination. Participation was on a voluntary basis therefore the survey had potential for self-selection bias by community members who are particularly concerned about the pandemic. However, probabilistic sampling (stratified Sampling method) was employed. The study also used self-administered questionnaires and the disadvantages of self-administered questionnaires is low response rates, exclusion of those who cannot read and write, and that the researcher cannot couch for the validity of the responses from self-administered surveys. However in our study the response rate was high more than 95% and also around 95.7% of participants had at least attended primary school level. Four point three (4.3%) percent of the participants did not respond the question on educational background. Furthermore, the investigators included all the nine COVID-19 zones of Botswana to ensure external validity. A total of 5300 out of 5745 people participated in the study, however an adequate response rate (92.2%) was achieved. The study was carried out at a time when people's perceptions may be highly volatile due to exposure to several other opinions on the internet. Replication of the study at different points during the course of vaccine roll-out will be beneficial. The study did not explore in depth the reasons for the high acceptance rate and low risk perception. A qualitative study is recommended to explore these reasons.

## Conclusions

This study revealed a high acceptance rate and low risk perception for COVID-19 vaccination. A strong communication plan is required to address the factors that affect acceptance rates and risk perception for a successful vaccination campaign.

## Supporting information

**S1 Data.**
(XLSX)

**S1 Questionnaire.**
(PDF)

**S2 Questionnaire.**
(PDF)

## Acknowledgments

The researchers would like to thank the following for their constructive feedback and valuable input on different sections of this paper. Ogomoditse Machinya, Kenosi Mogorosi, Motshidisi Mphotwe, Badubi Barcun, Mabedi Mogapi, Gogaone Setutu, Abigail Lesego Olesitse, Balekanye Boithatelo, Koziba Meshack, Lucrecia Moremi, Masego Nkwe, Nkgadimang Stegling, Pauline Mahilo,Pusetso Setshwantsho, Sylvester Pogiso, Kedirileng Ramakama, Dorcus Motsamai, Veronica Mogorosi, Doreen Motshegwa and Dziidzo Leshiba.

We also thank the FHI Botswana for also participating in this research project.

## Author Contributions

**Conceptualization:** Lebapotswe B. Tlale, Lesego Gabaitiri, Lorato K. Totolo, Gomolemo Smith, Eunice Ramonna, Basego Mothowaeng, Faith Mafa, Samuel Kolane.

**Data curation:** Lebapotswe B. Tlale, Lesego Gabaitiri, Lorato K. Totolo, Gomolemo Smith, Orapeleng Puswane-Katse, Eunice Ramonna, Basego Mothowaeng, John Tlhakanelo, Tiny Masupe, Goabaone Rankgoane-Pono, Samuel Kolane.

**Formal analysis:** Lebapotswe B. Tlale, Lesego Gabaitiri, Lorato K. Totolo, Gomolemo Smith, Orapeleng Puswane-Katse, Eunice Ramonna, Tiny Masupe, Goabaone Rankgoane-Pono, John Irige.

**Funding acquisition:** Lebapotswe B. Tlale, Gomolemo Smith, Orapeleng Puswane-Katse, Faith Mafa, Samuel Kolane.

**Investigation:** Lebapotswe B. Tlale, Lesego Gabaitiri, Lorato K. Totolo, Gomolemo Smith, Orapeleng Puswane-Katse, Basego Mothowaeng, Faith Mafa.

**Methodology:** Lebapotswe B. Tlale, Lesego Gabaitiri, Lorato K. Totolo, Eunice Ramonna, Basego Mothowaeng, John Tlhakanelo, Tiny Masupe, Goabaone Rankgoane-Pono, Samuel Kolane.

**Project administration:** Lebapotswe B. Tlale, Lesego Gabaitiri, Lorato K. Totolo, Gomolemo Smith, Orapeleng Puswane-Katse, Eunice Ramonna, Basego Mothowaeng, Faith Mafa.

**Resources:** Lebapotswe B. Tlale, Lesego Gabaitiri, Lorato K. Totolo, Orapeleng Puswane-Katse, Faith Mafa, Samuel Kolane.

**Software:** Lebapotswe B. Tlale, Lesego Gabaitiri, John Irige, Faith Mafa.

**Supervision:** Lebapotswe B. Tlale, Lesego Gabaitiri, Lorato K. Totolo, Orapeleng Puswane-Katse, Eunice Ramonna, Basego Mothowaeng, Tiny Masupe, Goabaone Rankgoane-Pono, Faith Mafa, Samuel Kolane.

**Validation:** Lebapotswe B. Tlale, Lesego Gabaitiri, Gomolemo Smith, Eunice Ramonna, John Tlhakanelo, Tiny Masupe, Goabaone Rankgoane-Pono, John Irige.

**Visualization:** Lebapotswe B. Tlale, Lorato K. Totolo, Gomolemo Smith, Goabaone Rankgoane-Pono, John Irige.

**Writing – original draft:** Lebapotswe B. Tlale, Lesego Gabaitiri, Gomolemo Smith, Orapeleng Puswane-Katse, Eunice Ramonna, Basego Mothowaeng, John Tlhakanelo, Tiny Masupe, Goabaone Rankgoane-Pono, John Irige, Faith Mafa, Samuel Kolane.

**Writing – review & editing:** Lebapotswe B. Tlale, Lesego Gabaitiri, Lorato K. Totolo, Gomolemo Smith, Orapeleng Puswane-Katse, Eunice Ramonna, Basego Mothowaeng, John Tlhakanelo, Tiny Masupe, Goabaone Rankgoane-Pono, John Irige, Faith Mafa, Samuel Kolane.

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
