## [Decision Letter · Decision Letter 0]

18 May 2021

PONE-D-21-09673

Acceptance Rate and Risk Perception towards the COVID-19 Vaccine in Botswana: A Cross-Sectional Study

PLOS ONE

Dear Dr. Tlale,

Thank you for submitting your manuscript to PLOS ONE. After careful consideration, we feel that it has merit but does not fully meet PLOS ONE’s publication criteria as it currently stands. Therefore, we invite you to submit a revised version of the manuscript that addresses the points raised during the review process.

We look forward to receiving your revised manuscript.

Kind regards,

David Teye Doku

Academic Editor

PLOS ONE

Additional Editor Comments:

Dear Tlale,

Thank you for the considering Plos One for your research output. The reviewers have now completed their evaluation of your manuscript. They have raise many major concerns which need to be addressed before the manuscript can be accepted for publication. Therefore, we would like to invite you to appropriately address all the suggestions raised by the reviewers and indicate the revisions made to in the manuscript in accordance with the journals guidelines for submission of revised manuscript. Please pay attention to the revision of the methods section to ensure that it can be understood by the international audience.

Journal Requirements:

3. Please include additional information regarding the survey or questionnaire used in the study and ensure that you have provided sufficient details that others could replicate the analyses. For instance, if you developed a questionnaire as part of this study and it is not under a copyright more restrictive than CC-BY, please include a copy, in both the original language and English, as Supporting Information. Moreover, please include more details on how the questionnaire was pre-tested, and whether it was validated.

[We also thank the FHI Botswana for partially funding this research project.]

 [The funders had no role in study design, data collection and analysis, decision to publish, or preparation of the manuscript.]

5. Thank you for stating the following in the Financial Disclosure section:

 [The funders had no role in study design, data collection and analysis, decision to publish, or preparation of the manuscript.]

We note that one or more of the authors are employed by a commercial company: FHI 360

7. We note you have included a table to which you do not refer in the text of your manuscript. Please ensure that you refer to Table 2, 7 and 8 in your text; if accepted, production will need this reference to link the reader to the Table.

8. Please include a title for Table 4.

Reviewers' comments:

Reviewer's Responses to Questions

**Comments to the Author**

1. Is the manuscript technically sound, and do the data support the conclusions?

Reviewer #1: Partly

Reviewer #2: Yes

Reviewer #3: Yes

Reviewer #4: Partly

2. Has the statistical analysis been performed appropriately and rigorously? 

Reviewer #1: No

Reviewer #2: No

Reviewer #3: I Don't Know

Reviewer #4: I Don't Know

3. Have the authors made all data underlying the findings in their manuscript fully available?

Reviewer #1: Yes

Reviewer #2: No

Reviewer #3: Yes

Reviewer #4: Yes

4. Is the manuscript presented in an intelligible fashion and written in standard English?

Reviewer #1: Yes

Reviewer #2: Yes

Reviewer #3: Yes

Reviewer #4: Yes

5. Review Comments to the Author

Reviewer #1: Thank you for considering me to review this manuscript, “Acceptance Rate and Risk Perception towards the COVID-19 Vaccine in Botswana: Across-Sectional Study”. As the only viable option to mitigate the COVID-19 pandemic is via vaccination, this implies understanding vaccination hesitancy is paramount so that factors that contribute to vaccination hesitancy could be recognized so that effectively counteract measures -health education - could be contemplated. This study purports to explore the acceptance rate and risk perception of COVID-19 vaccines amongst the general population in southern Africa country, Botswana. The authors developed their questions to solicit acceptance rate and risk perception. The authors also included various socio-demographic information and clinical risk factors. The result seems interesting but the authors need to address some issues to render their manuscript with high caliber scientific merit. Some comments for the authors are detailed below

TITLE

The content of the study -the instrument used- is currently not clear to the readers. I would think here that the title would once the outcome measures are made explicit in the method section.

ABSTRACT

The ABSTRACT in general has been structured according to the style of this journal. Minor issues: change “Introduction” to “Background”. During COVID, there is a ‘tradition’ in most journals to include the dates of the data collection (1-28 February 2021).

It seems there is the repetition of the aims (“This study, aims to assess the acceptance rate and risk perception of COVID-19 vaccines amongst the general population in Botswana” … “This study aims to assess the acceptance rate and risk perception of COVID-19 vaccines amongst the general population in Botswana”. SUGGESTION, keep one and create a subheading -Method. In the method, bring outcomes measures on board and the fact the study explored socio-demographic information and some risk factors.

This information is not needed (…” and 31.3% of participants”) (“At P<0.05, the following...”. Suggestion: delete

Word employed should be geared toward the outcome measures of the study.

KEYWORD. This keyword should be included (“vaccine hesitancy”)

BACKGROUND INFORMATION (INTRODUCTION)

The background information appears to lead the reader to sense what is coming up as the aims of this study -well done.

This statement should be rephrased (“Vaccine acceptance mirrors the public’s perception towards the disease threat, demand for and attitude towards the vaccine”)

The word ‘race’ (“With regards to race, African-Americans and Hispanics demonstrated higher vaccine hesitancy than other races in a US study [16].”) is confusing. Suggestion: ethic group or simple state as following (“There is subcultural diversity, African-Americans and Hispanics demonstrated higher vaccine hesitancy than other cultural or ethnic groups in a US study”.

To avoid too much rambling (“Global literature has also recognized the distinctive roles of several socio-demographic determinants of health in vaccine acceptance and risk perception. Additionally, ….Moreover, credible sources of information about vaccines such as government were reported to instil high levels of trust amongst the public[10].”). Since these paragraphs are preceded by a literature view on the rate of vaccine hesitancy, this paragraph should simply articulate ‘associated factors of vaccine hesitancy’. For a better flow of information, the narration on the rate of vaccine hesitancy should be sent to the previous paragraph.

For aims (“We therefore assessed the acceptance and perceptions of COVID-19

vaccines in Botswana in order to inform the planned population roll out of the vaccines.”), it should reflect the content of the instrument used or developed. Also, the associated factors explored should be mentioned as one of the aims.

MATERIALS AND METHODS

The problem with the METHOD is that the authors failed to elaborate on the development of the questionnaire used. Further, there is no information on the reliability and other important information on its applicability to the present population. Please attend to this issue.

Overall, the method has much to be desired. I would encourage the authors to stick with the subheadings that are recommended by Strobe observational study checklist (https://www.strobe-statement.org/index.php?id=available-checklists).

This statement touches on an important part of the study (“Following consent, participants were enrolled into the study by administering the questionnaires in the selected localities. This happened at government departments, shops, markets and private companies”.). How data collection was done should be more explicit in the text. Please identify the name of the technique for data collection (probabilistic sampling as stated in the limitation? ).

Different parts of the text describing recruiting and data collection should be consistent with each other. Thus, strict adherence to Strobe observational study checklist would help.

For us international reader, we may not understand what constitute “across the nine (9) COVID-19 zones in Botswana”. Please define.

The rationale for including this statement (“A risk perception model which integrates three core dimensions; cognitive factors (knowledge), experiential factors (emotion) and socio-cultural factors (norms, values) was used to guide in the development of the questions”) is needed. Also, a separate section is needed to highlight the development of the questionnaire. This should be merged with this (“designed from literature review [12, 21, 22]”).

Any reference number was assigned for the present ethical approval from IRB?

The result section has a lot of information and some of them were not described in the methodology or stated as the aim of the study. This is major misgiving of this manuscript. The confusion partly stems from the factor that outcome measures were not described in the method.

Information in different tables should be merged into coherent themes.

DISCUSSION

I am not going to dwell too much on the discussion section since would invariably be changed if the above-mentioned suggestions are contemplated

ACKNOWLEDGEMENT

You want to separate acknowledged name (ACKNOWLEDGMENT) person by coma rather than bullets.

REFERENCES

The authors have employed 30 references. Most of them are relevant and update.

Reviewer #2: General Comments

This is a relevant and current topic. As countries work to start COVID-19 vaccination programmes, the issues of acceptance and risk perception are essential research topics. This is more so in Africa where the epidemiology of COVID-19 has been different and thus risk perception and acceptance of vaccination is likely to differ from developed countries. The manuscript is generally well written.

Specific Comments:

1. Throughout the manuscript, you use COVID at some places and COVID-19 at other places. Please use COVID-19.

2. Title: “a cross–sectional study” in my opinion does not add anything useful to the title while increasing the length. So my suggestion is to remove that

3. Abstract: there is repetition of the study objective at the end of the introduction and then the objective parts. I suggest you remove it at the end of the introduction part

4. At the results part of the abstract you have “At P<0.05, the following factors were associated with willingness to take the COVID vaccine: gender, education level, occupation and COVID zone”.

To make it more meaningful in my opinion, kindly consider revising as ‘factors found to be associated with willingness to take the COVID vaccine were gender (p=…..), education level (p=…), occupation (p=…) and COVID zone (p=…..)’. And do put the actual p values

5. Then you have “Safety of the vaccine was associated with age group and religion”. Give readers the evidence of the association please, state the p values at least.

6. You also have “the acceptance rate of COVID-19 vaccine in Botswana was 73.4% and 31.3% of participants perceived the COVID vaccine as unsafe”. What was the 95% confidence intervals for these rates? That will be very useful

7. Introduction: generally I feel this could be shorter than current length. It has some aspects more sounding likely discussion using many studies from other places. That could be made more concise and therefore reduce the length. Consider this.

8. Study population: you just mentioned that stratified sampling was used to select study participants but offer on more details. How was this done exactly? Were the population size for each of the zones factored into the number recruited per zone? You do not tell us how the 9 zones were created and how that related to the study design.

9. Then you say “Individuals/study participants were selected within the localities randomly selected from the Covid-19 zones”, how was this done?

10. Was there any exclusion criteria

11. You have “To cover the 9 COVID-19 zones, we estimated a sample of 383 multiplied by giving a total of 3447”, there is something missing. It does not read well to me, kindly have a look.

12. Since this was in-person interviewer administered questionnaire, how was the safety of everyone involved assured?

13. Results: if this was interviewer administered why do you have so many missing values for almost each of the variables?

14. Consider presenting table 2 and 4 as figures rather, I think they will be nicer e.g. as a pie chart or other appropriate figure type. And most importantly, please indicate the 95%CI for the acceptance rates

15. You have just too many tables. Apart from those I have pointed out can be figures, there are still a number of tables that can be put together as 1 larger table and will still be clear. Have a look and do this for tables 6-8 and then 9-10.

16. You mentioned multivariate analysis but I see no such results presented

17. Check and deal with some few long sentences and grammatical errors.

Reviewer #3: I have some comments about the manuscript regarding the content of tables, result part, and some minor language issues that follow below. I would strongly recommend that the authors recheck the numbers in the tables.

Results

1)It seems that an error exists in Table 1, page 11 regarding total number of “education level” which is 5310?! Why not 5300?

2) In Table 2, page 12: why total number is not 5300? If missing data exist, I suggest that the authors add them to the table

3) Page 12, the part d of the results about “Factors associated with acceptance for the COVID vaccine in Botswana”:

as all the factors have been explained, a short explanation about how occupations were associated with acceptance of the COVID vaccine is good to be added to the text

4) Regarding Table 4, page 13, as total number is 4784, whether 516 is missing here? Then it is better to provide info about missing data in the table

5) Regarding Table 6, page 14, are there missing data? then it is better to provide the number of missing data in the table

6) Regarding Table 7, page 15, I think it is better to add info about missing data if exist

7) In Table 8: it seems that an error exists regarding total numbers: 1172+3772

8) In Table 9, page 16: it seems that an error exists here. Regarding total frequencies, why is it 5931?

and percent which is 111,9 ?!

9) Table 10: It seems that an error exists here too regarding total frequencies:

why is it 6203? and percent which is 117 ?!

*And some minor language issues:

page 15 line 2: Forty nine percent of participants who said their religion and “cultural” hinders vaccine uptake were willing to receive the COVID-19 vaccine.

use the word "culture" instead of cultural

Regarding title of Table 7, delete the first “believes” and add "religious" instead of religion

Page 15: regarding the line that follows “Willing to receive COVID vaccine though trust other traditional and religious methods over vaccine”

delete i

Reviewer #4: Thank you for considering me to review this manuscript, “Acceptance Rate and Risk Perception towards the COVID-19 Vaccine in Botswana: Across-Sectional Study”. As the only viable option to mitigate the COVID-19 pandemic is via vaccination, this implies understanding vaccination hesitancy is paramount so that factors that contribute to vaccination hesitancy could be recognized so that effectively counteract measures -health education - could be contemplated. This study purports to explore the acceptance rate and risk perception of COVID-19 vaccines amongst the general population in southern Africa country, Botswana. The authors developed their questions to solicit acceptance rate and risk perception. The authors also included various socio-demographic information and clinical risk factors. The result seems interesting but the authors need to address some issues to render their manuscript with high caliber scientific merit. Some comments for the authors are detailed below

TITLE

The content of the study -the instrument used- is currently not clear to the readers. I would think here that the title would once the outcome measures are made explicit in the method section.

ABSTRACT

The ABSTRACT in general has been structured according to the style of this journal. Minor issues: change “Introduction” to “Background”. During COVID, there is a ‘tradition’ in most journals to include the dates of the data collection (1-28 February 2021).

It seems there is the repetition of the aims (“This study, aims to assess the acceptance rate and risk perception of COVID-19 vaccines amongst the general population in Botswana” … “This study aims to assess the acceptance rate and risk perception of COVID-19 vaccines amongst the general population in Botswana”. SUGGESTION, keep one and create a subheading -Method. In the method, bring outcomes measures on board and the fact the study explored socio-demographic information and some risk factors.

This information is not needed (…” and 31.3% of participants”) (“At P<0.05, the following...”. Suggestion: delete

Word employed should be geared toward the outcome measures of the study.

KEYWORD. This keyword should be included (“vaccine hesitancy”)

BACKGROUND INFORMATION (INTRODUCTION)

The background information appears to lead the reader to sense what is coming up as the aims of this study -well done.

This statement should be rephrased (“Vaccine acceptance mirrors the public’s perception towards the disease threat, demand for and attitude towards the vaccine”)

The word ‘race’ (“With regards to race, African-Americans and Hispanics demonstrated higher vaccine hesitancy than other races in a US study [16].”) is confusing. Suggestion: ethic group or simple state as following (“There is subcultural diversity, African-Americans and Hispanics demonstrated higher vaccine hesitancy than other cultural or ethnic groups in a US study”.

To avoid too much rambling (“Global literature has also recognized the distinctive roles of several socio-demographic determinants of health in vaccine acceptance and risk perception. Additionally, ….Moreover, credible sources of information about vaccines such as government were reported to instil high levels of trust amongst the public[10].”). Since these paragraphs are preceded by a literature view on the rate of vaccine hesitancy, this paragraph should simply articulate ‘associated factors of vaccine hesitancy’. For a better flow of information, the narration on the rate of vaccine hesitancy should be sent to the previous paragraph.

For aims (“We therefore assessed the acceptance and perceptions of COVID-19

vaccines in Botswana in order to inform the planned population roll out of the vaccines.”), it should reflect the content of the instrument used or developed. Also, the associated factors explored should be mentioned as one of the aims.

MATERIALS AND METHODS

The problem with the METHOD is that the authors failed to elaborate on the development of the questionnaire used. Further, there is no information on the reliability and other important information on its applicability to the present population. Please attend to this issue.

Overall, the method has much to be desired. I would encourage the authors to stick with the subheadings that are recommended by Strobe observational study checklist (https://www.strobe-statement.org/index.php?id=available-checklists).

This statement touches on an important part of the study (“Following consent, participants were enrolled into the study by administering the questionnaires in the selected localities. This happened at government departments, shops, markets and private companies”.). How data collection was done should be more explicit in the text. Please identify the name of the technique for data collection (probabilistic sampling as stated in the limitation? ).

Different parts of the text describing recruiting and data collection should be consistent with each other. Thus, strict adherence to Strobe observational study checklist would help.

For us international reader, we may not understand what constitute “across the nine (9) COVID-19 zones in Botswana”. Please define.

The rationale for including this statement (“A risk perception model which integrates three core dimensions; cognitive factors (knowledge), experiential factors (emotion) and socio-cultural factors (norms, values) was used to guide in the development of the questions”) is needed. Also, a separate section is needed to highlight the development of the questionnaire. This should be merged with this (“designed from literature review [12, 21, 22]”).

Any reference number was assigned for the present ethical approval from IRB?

The result section has a lot of information and some of them were not described in the methodology or stated as the aim of the study. This is major misgiving of this manuscript. The confusion partly stems from the factor that outcome measures were not described in the method.

Information in different tables should be merged into coherent themes.

DISCUSSION

I am not going to dwell too much on the discussion section since would invariably be changed if the above-mentioned suggestions are contemplated

ACKNOWLEDGEMENT

You want to separate acknowledged name (ACKNOWLEDGMENT) person by coma rather than bullets.

REFERENCES

The authors have employed 30 references. Most of them are relevant and update.

6. PLOS authors have the option to publish the peer review history of their article (what does this mean?). If published, this will include your full peer review and any attached files.

Reviewer #1: **Yes: **Samir Al-Adawi

Reviewer #2: No

Reviewer #3: No

Reviewer #4: **Yes: **Samir Al-Adawi

---

## [Author Response · Author response to Decision Letter 0]

16 Jul 2021

Dr Lebapotswe Bahumi Tlale, MBBS

 16th June 2021

David Teye Doku

Academic Editor

PLOS ONE

Dear Sir

RE: Manuscript ID PONE-D-21-09673 Acceptance Rate and Risk Perception towards the COVID-19 Vaccine in Botswana: A Cross-Sectional Study. PLOS ONE

Thank you for your and the reviewer’s thoughtful review of our manuscript. With these in mind we are pleased to submit a revised manuscript for publication in the PLOS ONE Journal. The manuscript has not been published, and is not being considered for publication elsewhere.

These are our responses to each point raised by the academic editor and the reviewers:

Academic Editors Comments:

Comment 1: Please ensure that your manuscript meets PLOS ONE's style requirements, including those for file naming. The PLOS ONE style templates can be found at

Response: Thank you for clarifying this to us. We have formatted the manuscript to meet the PLOS ONE style requirements as advised by the academic editor.

Comment 2: Please amend your current ethics statement to include the full name of the ethics committee/institutional review board(s) that approved your specific study.

Response: Thank you for highlighting this to us. We have included the following statement in our manuscript. The protocol and any amendments made were submitted to the Botswana, Ministry of Health and Wellness an independent Ethics Committee (EC) or Institutional Review Board (IRB), were it was approved”. We have also attached the approval certificate as part of documents submitted.

Comment 3: Please include additional information regarding the survey or questionnaire used in the study and ensure that you have provided sufficient details that others could replicate the analyses. For instance, if you developed a questionnaire as part of this study and it is not under a copyright more restrictive than CC-BY, please include a copy, in both the original language and English, as Supporting Information. Moreover, please include more details on how the questionnaire was pre-tested, and whether it was validated.

Response: Thank you for your comment. We will include the copies of the questionnaires both in English and Original language as part of other documents submitted with the manuscript.

Comment 4: Thank you for stating the following in the Acknowledgments Section of your manuscript:

[We also thank the FHI Botswana for partially funding this research project.]

[The funders had no role in study design, data collection and analysis, decision to publish, or preparation of the manuscript.]

Response: Thank you for highlighting this to us and we have removed the statement from the acknowledgement section to read “We also thank the FHI Botswana for also participating in this research project.” We do not wish to change the funding statement and should remain as is. The partial funding that we referred to was payment of publication fees by FHI. They did not fund the research project.

FHI Botswana paid the article publication fee. No additional external funding was received for this study.

Comment 5: Thank you for stating the following in the Financial Disclosure section:

 [The funders had no role in study design, data collection and analysis, decision to publish, or preparation of the manuscript.]

We note that one or more of the authors are employed by a commercial company: FHI 360

Please know it is PLOS ONE policy for corresponding authors to declare, on behalf of all authors, all potential competing interests for the purposes of transparency. PLOS defines a competing interest as anything that interferes with, or could reasonably be perceived as interfering with, the full and objective presentation, peer review, editorial decision-making, or publication of research or non-research articles submitted to one of the journals. Competing interests can be financial or non-financial, professional, or personal. Competing interests can arise in relationship to an organization or another person. Please follow this link to our website for more details on competing interests: http://journals.plos.org/plosone/s/competing-interests.

Response: Thank you raising this concern. We would like to clearly state that there are no competing interest to declare in this research project. FHI as a commercial organisation is a member of the ministry of health and wellness communication team (Part of the National Public Health Emergency preparedness committee). The author or FHI did not fund and did not play a role in the study design, data collection and analysis, decision to publish, or preparation of the manuscript and also did not provide financial support in the form of authors' salaries and/or research materials for this study. The Author referred above was part of the manuscript write up and did not influence the results, discussion or conclusion of the study.

There was no funding received for this research project and none of the authors received any salaries for undertaking this research project. This applies to all the authors listed in the manuscript.

Comment 6: In your Data Availability statement, you have not specified where the minimal data set underlying the results described in your manuscript can be found. PLOS defines a study's minimal data set as the underlying data used to reach the conclusions drawn in the manuscript and any additional data required to replicate the reported study findings in their entirety. All PLOS journals require that the minimal data set be made fully available. For more information about our data policy, please see http://journals.plos.org/plosone/s/data-availability.

Response: Thank you for highlighting this to us. We will submit our minimal data set with this manuscript as a supporting information file.

Comment 7: We note you have included a table to which you do not refer in the text of your manuscript. Please ensure that you refer to Table 2, 7 and 8 in your text; if accepted, production will need this reference to link the reader to the Table.

Response: Thank you for bringing this to our attention. We have added some text to describe the above tables

Comment 8: Please include a title for Table 4.

Response: Thank you for bringing this to our attention. We have included a title for table 4.

Reviewer’s comments:

Reviewer 1 comments: Thank you for considering me to review this manuscript, “Acceptance Rate and Risk Perception towards the COVID-19 Vaccine in Botswana: Across-Sectional Study”. As the only viable option to mitigate the COVID-19 pandemic is via vaccination, this implies understanding vaccination hesitancy is paramount so that factors that contribute to vaccination hesitancy could be recognized so that effectively counteract measures -health education - could be contemplated. This study purports to explore the acceptance rate and risk perception of COVID-19 vaccines amongst the general population in southern Africa country, Botswana. The authors developed their questions to solicit acceptance rate and risk perception. The authors also included various socio-demographic information and clinical risk factors. The result seems interesting but the authors need to address some issues to render their manuscript with high calibre scientific merit.

Response: Thank you for the insights. We will work on the feedback and respond appropriately.

Comment •TITLE: The content of the study -the instrument used- is currently not clear to the readers. I would think here that the title would once the outcome measures are made explicit in the method section.

Response: Thank you for highlighting this to us. We have worked on our title and instrument used by responding appropriately to the comments made on the method section.

Comment •ABSTRACT: The ABSTRACT in general has been structured according to the style of this journal. Minor issues: change “Introduction” to “Background”. During COVID, there is a ‘tradition’ in most journals to include the dates of the data collection (1-28 February 2021).

It seems there is the repetition of the aims (“This study, aims to assess the acceptance rate and risk perception of COVID-19 vaccines amongst the general population in Botswana” … “This study aims to assess the acceptance rate and risk perception of COVID-19 vaccines amongst the general population in Botswana”. SUGGESTION, keep one and create a subheading -Method. In the method, bring outcomes measures on board and the fact the study explored socio-demographic information and some risk factors.

This information is not needed (…” and 31.3% of participants”) (“At P<0.05, the following...”. Suggestion: delete

Word employed should be geared toward the outcome measures of the study.

KEYWORD. This keyword should be included (“vaccine hesitancy”)

Response: The comments are all welcome and have been incorporate into the manuscript text. The abstract now reads as follows:

“Background: The COVID-19 disease burden continues to be high worldwide and vaccines continue to be developed to help combat the pandemic. Acceptance and risk perception for COVID-19 vaccines is unknown in Botswana despite the government’s decision to roll out the vaccine nationally. 

Objectives: This study aims to assess the acceptance rate and risk perception of COVID-19 vaccines amongst the general population in Botswana. 

Methods: We interviewed 5300 adults in Botswana from 1-28 February 2021 using self-administered questionnaires. The main outcomes of the study were vaccine acceptance and hesitancy rates. Demographic, experiential and socio-cultural factors were explored for their association with outcome variables.

Results: Two-thirds of the participants were females (3199), with those aged 24-54 making the highest proportion (61%). The acceptance rate of COVID-19 vaccine was 73.4% with vaccine hesitancy at 31.3%. Factors found to be associated with willingness to take the COVID-19 vaccine were sex (p= 0.033), age group (p= 0.009), education level (p= 0.000), willingness to wear face mask (p=0.000), religious beliefs (p=0.000), occupation (p=0.023) and marital status (p=0.045). Safety perception of the vaccine was associated with education level (p= 0.000), willingness to wear face mask (p=0.000), age group (p=0.000) and religion (p=0.001). Ninety six percent (96%) of those willing to receive the vaccine indicated willingness to continue wearing masks and social distancing after vaccination. Radio and Television were the preferred sources for COVID-19 prevention and vaccination information. 

Conclusion: This study demonstrated a high acceptance rate for the COVID-19 vaccine and a low risk perception in Botswana. In order to achieve a high vaccine coverage and ensure a successful vaccination process, there is need to target populations with high vaccine hesitancy rates. A qualitative study to assess the factors associated with vaccine acceptance and hesitancy is recommended to provide an in-depth analysis of the findings.”

 KEYWORDS: Vaccine hesitancy added to the key words. Keywords now read;

“Acceptance Rate, vaccine hesitancy, Risk Perception COVID-19 Vaccine, Case, Contact and Contact Tracing””

Comment: •BACKGROUND INFORMATION (INTRODUCTION)

The background information appears to lead the reader to sense what is coming up as the aims of this study -well done.

This statement should be rephrased (“Vaccine acceptance mirrors the public’s perception towards the disease threat, demand for and attitude towards the vaccine”)

The word ‘race’ (“With regards to race, African-Americans and Hispanics demonstrated higher vaccine hesitancy than other races in a US study [16].”) is confusing. Suggestion: ethic group or simple state as following (“There is subcultural diversity, African-Americans and Hispanics demonstrated higher vaccine hesitancy than other cultural or ethnic groups in a US study”.

To avoid too much rambling (“Global literature has also recognized the distinctive roles of several socio-demographic determinants of health in vaccine acceptance and risk perception. Additionally, ….Moreover, credible sources of information about vaccines such as government were reported to instil high levels of trust amongst the public[10].”). Since these paragraphs are preceded by a literature view on the rate of vaccine hesitancy, this paragraph should simply articulate ‘associated factors of vaccine hesitancy’. For a better flow of information, the narration on the rate of vaccine hesitancy should be sent to the previous paragraph.

For aims (“We therefore assessed the acceptance and perceptions of COVID-19 vaccines in Botswana in order to inform the planned population roll out of the vaccines.”), it should reflect the content of the instrument used or developed. Also, the associated factors explored should be mentioned as one of the aims.

Response: Thank you for this suggestion. We have amended the background as suggested. It now reads:

“ Background

According to the World Health Organisation (WHO) weekly epi update of the 14th February 2021, there were a total of 108 246 992 COVID-19 cases and 2 386 717 COVID-19 deaths [1]. Of these, the African region reported a total of 2 723 431 COVID-19 cases and 68 294 deaths, with Botswana reporting 24 926 total cases and 202 total deaths [1]. Vaccines are considered one of the most awaited interventions for combating COVID-19 and hundreds of global institutions are working at an unprecedented speed to develop COVID-19 vaccines [2-9]. Several vaccines have been developed and some are still undergoing clinical trials while very few countries have started vaccine rollout [10]. One of the challenges towards COVID-19 vaccination is the uncertainty of vaccine acceptance among the public. In general factors that influence vaccine acceptance include the public’s demand for vaccine, their perception towards the disease and attitudes towards the vaccine [11]

Acceptance and perception studies are important as they provide critical information for health education programs to increase uptake of vaccines and target certain populations. Few studies have been conducted assessing the acceptance, perceptions and attitudes towards COVID-19 vaccines as well as factors influencing uptake which may vary from country to country. A global survey of potential acceptance of a COVID-19 vaccine has shown that differences in acceptance rates ranged from almost 90% (in China) to less than 55% (in Russia) [10].In an Australian study [12], eighty per cent (80%) respondents generally held positive views towards COVID-19 vaccination while 65% of participants in Saudi Arabia [2], showed interest to accept the COVID-19 vaccine if available. The preliminary results of a Chilean case study on COVID-19 vaccine perception in the country showed that 87% were willing to vaccinate, a relatively high proportion and slightly lower than the rate found by García and Cerda [13] which was 90.6% for Chile. Gender and age disparities in attitudes were characterised in Australia, where females (83%) were found to be more likely to depict an optimistic outlook to receiving vaccinations than males (78%), as well as those aged 70 years and above (91%) compared to 76% of 18–29-year-olds [12]. Furthermore, willingness to accept future COVID-19 vaccines in Saudi Arabia was relatively high among older age groups, married participants, participants with a postgraduate degree or higher education level (68.8%), non-Saudi (69.1%), employed in government sector (68.9%) [2]. There is subcultural diversity as African Americans and Hispanics demonstrated higher vaccine hesitancy than other cultural or ethnic groups in a US study [16].

Religion, female gender, residing in deprived neighbourhoods were some factors found to be correlated with COVID-19 vaccine hesitancy in Australia (Edwards et al, 2021), while reliance on social media and refraining from news (Chadwick et al, 2021) were also associated with vaccine hesitancy in the UK.

These studies demonstrate significant variations in COVID-19 vaccine acceptance across different countries and the roles of several socio-demographic determinants of health in vaccine acceptance and risk perception. Such information is critical for context and country specific implementation of COVID-19 vaccination programs, with governments being encouraged to understand communities’ concerns and identify strategies that will support engagement to support effective launching of new vaccines [12]

Circumstantial conditions such as the pandemic context, specifically disease prevalence in a particular population can also impact vaccination intention [13]. Perceived severity of COVID-19 and perceived vaccine safety were the two strongest determinants of vaccine acceptance in a Finland study [14]. In Turkey and the UK, acceptance rates of vaccines were found to be higher among study participants who believed in the natural origin of COVID-19 in contrast to those who believed that the disease was generated by humans [15]. Moreover, credible sources of information about vaccines such as government were reported to instil high levels of trust amongst the public [10]. 

In the African setting, the level of vaccine acceptance (53.6%) and risk perception of (46.7%) were relatively average in Western Uganda [17]. Males, those with a tertiary education, students and non-salary earners were likely to accept the vaccine [17]. 

The Botswana government has adopted a multi-pronged strategy as part of the response to COVID-19. One of the pillars of the COVID-19 epidemic control measures includes vaccinating 276, 078 (16.5%) of the targeted population in the 1st phase of the vaccination campaign. Botswana is a signatory to the WHO/Worldbank vaccine initiative and expects to receive its first COVID-19 vaccines in March 2021. Acceptance and risk perception for COVID-19 vaccines is unknown in Botswana. There have been several conspiracy theories around COVID-19 vaccines which are mostly linked to religious and cultural beliefs which may influence the uptake of the vaccine [19]. The widely used social media platforms have also reported on the negative aspects of the vaccine and its side effects which may also influence the population’s attitudes and perceptions resulting in low acceptance [20]. We therefore assessed the acceptance and risk perceptions of COVID-19 vaccines in Botswana in order to inform the planned population roll out of the vaccines. Demographic, experiential and socio-cultural factors were also explored for their association with the outcome measures.”

Comment •MATERIALS AND METHODS: The problem with the METHOD is that the authors failed to elaborate on the development of the questionnaire used. Further, there is no information on the reliability and other important information on its applicability to the present population. Please attend to this issue.

Overall, the method has much to be desired. I would encourage the authors to stick with the subheadings that are recommended by Strobe observational study checklist (https://www.strobe-statement.org/index.php?id=available-checklists).

This statement touches on an important part of the study (“Following consent, participants were enrolled into the study by administering the questionnaires in the selected localities. This happened at government departments, shops, markets and private companies”.). How data collection was done should be more explicit in the text. Please identify the name of the technique for data collection (probabilistic sampling as stated in the limitation? ).

Different parts of the text describing recruiting and data collection should be consistent with each other. Thus, strict adherence to Strobe observational study checklist would help.

For us international reader, we may not understand what constitute “across the nine (9) COVID-19 zones in Botswana”. Please define.

The rationale for including this statement (“A risk perception model which integrates three core dimensions; cognitive factors (knowledge), experiential factors (emotion) and socio-cultural factors (norms, values) was used to guide in the development of the questions”) is needed. Also, a separate section is needed to highlight the development of the questionnaire. This should be merged with this (“designed from literature review [12, 21, 22]”).

Any reference number was assigned for the present ethical approval from IRB?

The result section has a lot of information and some of them were not described in the methodology or stated as the aim of the study. This is major misgiving of this manuscript. The confusion partly stems from the factor that outcome measures were not described in the method.

Information in different tables should be merged into coherent themes.

Response: Thank you for clarifying this to us. We reviewed our methodology section and used the strobe guidelines to write our methods section clearly. The new method section is as below.

“Materials and Methods

Study Design

This study used a cross sectional survey design conducted across the nine (9) COVID-19 zones in Botswana from 1-28 February 2021 using enumerator administered questionnaires designed from literature review and using a risk perception model which integrates three core dimensions; cognitive factors (knowledge), experiential factors (emotion) and socio-cultural factors (norms, values) [12, 23, 24]. The questionnaires were self-administered without any interference from the interviewer. 

Study Setting 

Botswana is a landlocked country situated in Southern Africa and shares borders with South Africa, Namibia, Zimbabwe and Zambia. The dynamics of population movement across borders with neighbouring countries increases the risk of transmission of communicable diseases including COVID-19. Botswana has an estimated population of 2.374,697, (1,171,629 Males and 1,203,068 Females) as per the 2011 population census projection for 2020 [22]. The population is spread over vast land of 581,730km2. The estimated population density in Botswana is 4 per Km. Sixty nine percent (69.4%) of the total population lives in urban settings.

Botswana’s economy is largely dependent on mineral revenues and belongs to Southern Africa Custom Union. The Gross National Income (GNI) per capita is approximately $6,000, thus by classification, Botswana belongs to the upper middle-income countries. Botswana recorded its first COVID-19 case in March 2020. After recording the first case the country was demarcated into nine (9) sections called COVID-19 zones in order to restrict movement of non-essential travel between these zones. In order to travel between these zones, a valid permit, which should be applied for online, is necessary. 

Participants 

The study population consisted of 5300 adults aged 18 years and above since most candidate vaccines were investigated and tried in this group worldwide. In addition, this was in line with the Botswana government policy for COVID-19 vaccine eligibility. Study participants were selected from localities in each of the nine (9) Botswana COVID-19 zones using stratified sampling. Eligibility for participation in the study included those aged 18 years and above including pregnant women, and those with (and without) comorbidities and able to give informed consent. Following consent, participants were enrolled into the study by administering the questionnaires in the selected localities. This happened at government departments, shops, markets and private companies. Exclusion criteria included those who are under 18 years of age and those that did not consent to participate in the study.

Study Size

The survey leveraged on the already existing resources. The nine (9) COVID-19 zones were used as strata (Table 1). A stratified sampling was employed for this survey. Individuals/study participants were selected within the localities randomly selected from the COVID-19 zones. 

To compute the sample size for the survey a margin of error of 0.05 and a confidence level of 0.95 were used. We further assumed an acceptance rate of 53.6% which is similar to the one estimated by Echoru I et al (2020) in Uganda [19] resulting in a sample of size 383. To cover the 9 COVID-19 zones, we estimated a sample size of 383 multiplied by nine (9) zones giving a total of 3447. To accommodate the design effect, since our sampling strategy is stratified sampling scheme, we assumed a design effect of 1.5 which then gave us a sample size of 5171. Finally, we assumed 90% response rate which resulted in a total sample of 5745 for the survey. Table 1. The 9 COVID-19 Zones, Sample Size and Allocation

Zone Population Proportional Sample

 Localities >=18 years Locality Individuals

Greater Gaborone 167 590,689 3 2712

Greater Palapye 78 188,428 2 865

Maun 52 84,449 1 388

Greater Francistown 80 192,483 2 884

Chobe 18 23,789 1 109

Ghanzi 17 22,474 1 103

Kgalagadi 47 30,678 1 141

Selebi Phikwe 21 74,678 1 343

Boteti 20 43,810 1 201

Total 500 1,238,768 13 5745

Variables

The exposure variables for this study included age, gender, level of education, Occupation, religious background, Media, marital status and place of residence. The primary outcome measure includes acceptance rate and risk perception towards the COVID-19 vaccine.

Bias

Participation was ere selected on a voluntary basis therefore the survey had potential for self-selection bias by community members who are particularly concerned about the pandemic. However, probabilistic sampling was employed. The study also used self-administered questionnaires and this could have an impact on the response rate by the participants. 

Data Sources

The cross-sectional survey used questionnaires to interview participants from selected areas across the country. The questionnaires were written in both English and Setswana and data obtained from the questionnaire were analysed using the STATA statistical software version 15. The questionnaires were validated before being administered to all the participants. The questionnaires were The participants answered the same questionnaire across zones to obtain reliable information. Data protection was ensured via file password protection and limiting access of information to the project team members only. 

Statistical Methods

Categorical variables were analysed using summary statistics such as frequencies and percentages. For continuous variables such as age, median (inter-quartile range) or mean (and standard deviation) were presented depending on whether the variable under consideration is skewed or not, respectively. Acceptance was measured by the question asking respondents if they are willing to receive the COVID-19 vaccine when it is rolled out nationally. The acceptance rate of COVID-19 vaccine was the number that “accept” divided by the sample size but excluding those with missing responses. The independent variables were demographic variables such as age, gender, marital status, educational level, and other important variables such presence and absence of comorbidities, COVID-19 zone level (high/red zone, medium, low), socio-cultural and experimental factors. Risk perception was measured by the questions asking respondents if they considered the COVID-19 vaccine to be safe.

Bivariate analysis using the chi-square test was conducted to determine factors associated with both acceptance and risk perception for the COVID-19 vaccine. Multiple logistic regression was conducted to adjust for any confounding factors. Odds ratios, as measures of effect, and the corresponding p-values and 95% confidence intervals are presented. A p-value less than 0.05 was used to determine significance.

Ethical considerations

This study was conducted according to Botswana, and International Standards of Good Clinical Practice, applicable government regulations and Institutional research policies and procedures. The protocol and any amendments made were submitted to the Botswana, Ministry of Health and Wellness Institutional Review Board (IRB), were it was approved. The protocol number awarded for this study is HPDME 13/18/1 

All subjects were asked to provide written informed consent before responding to questions and they were free to withdraw at any time during the study.”

•DISCUSSION I am not going to dwell too much on the discussion section since would invariably be changed if the above-mentioned suggestions are contemplated

Response: Thank you for highlighting this to us we have worked on our discussion section to be aligned to the methods and objectives of the study. 

•ACKNOWLEDGEMENT 

You want to separate acknowledged name (ACKNOWLEDGMENT) person by coma rather than bullets

Response: Thank you for highlighting this to us. We have corrected this mistake in the Acknowledgement section

•REFERENCES

The authors have employed 30 references. Most of them are relevant and update.

Response: Thank you for this comment. We will continue to use relevant and updated literature.

Reviewer 2 comments: 

Comment 1. General Comments

This is a relevant and current topic. As countries work to start COVID-19 vaccination programmes, the issues of acceptance and risk perception are essential research topics. This is more so in Africa where the epidemiology of COVID-19 has been different and thus risk perception and acceptance of vaccination is likely to differ from developed countries. The manuscript is generally well written.

Response: Thank you for the insights. We will work on the feedback and respond appropriately.

Specific Comments:

1. Throughout the manuscript, you use COVID at some places and COVID-19 at other places. Please use COVID-19.

Response: Thank you for this clarification. We have revised the whole manuscript and corrected this mistake. 

2. Title: “a cross–sectional study” in my opinion does not add anything useful to the title while increasing the length. So my suggestion is to remove that

Response: Thank you for highlighting this to us. “a cross–sectional study” removed from title: It now reads; Acceptance Rate and Risk Perception towards the COVID-19 Vaccine in Botswana

3. Abstract: there is repetition of the study objective at the end of the introduction and then the objective parts. I suggest you remove it at the end of the introduction part

Response: Thank you for this clarification. The aim has been removed from the end of the introduction part and placed appropriately as advised.

4. At the results part of the abstract you have “At P<0.05, the following factors were associated with willingness to take the COVID vaccine: gender, education level, occupation and COVID zone”.To make it more meaningful in my opinion, kindly consider revising as ‘factors found to be associated with willingness to take the COVID vaccine were gender (p=…..), education level (p=…), occupation (p=…) and COVID zone (p=…..)’. And do put the actual p values

Response: Thank you for this clarification. The statement has been revised as suggested and the actual p-values included. It now reads:

“Factors found to be associated with willingness to take the COVID vaccine were gender (p= 0.033), age group (p= 0.009), education level (p= 0.000), willingness to wear face mask (p=0.000) and Religious beliefs (p=0.000)”

5. Then you have “Safety of the vaccine was associated with age group and religion”. Give readers the evidence of the association please, state the p values at least.

Response: Thank you for this significant in-sight. The p values have been stated as suggested. The statement now reads:

“Safety perception of the vaccine was associated with education level (p= 0.000), willingness to wear face mask (p=0.000) age (p=0.000) and religion (p=0.001)”

6. You also have “the acceptance rate of COVID-19 vaccine in Botswana was 73.4% and 31.3% of participants perceived the COVID vaccine as unsafe”. What was the 95% confidence intervals for these rates? That will be very useful

Response: Thank you for highlighting this to us. The Confidence intervals have been added to the manuscript and the tables converted to pie charts as advised by the reviewer in other comments made to this document. The document now reads: 

“Figure 1 shows that out of 5 027 participants, 3689 were willing to take the vaccine resulting in the acceptance rate of the COVID-19 vaccine in Botswana of 73.4% ( CI: 72.2%, 4.6%)

Figure 2 below shows that out of 4784 participants, 1499 (31.3%) (CI: 30.0%, 32.6%) of participants believed that the COVID-19 vaccine was not safe.”

7. Introduction: generally I feel this could be shorter than current length. It has some aspects more sounding likely discussion using many studies from other places. That could be made more concise and therefore reduce the length. Consider this.

Response: Thank you for this clarification. Some of the statements have been removed from the introduction, while some were compressed to reduce the length and improve the flow. The background now reads:

“Background

According to the World Health Organisation (WHO) weekly epi update of the 14th February 2021, there were a total of 108 246 992 COVID-19 cases and 2 386 717 COVID-19 deaths [1]. Of these, the African region reported a total of 2 723 431 COVID-19 cases and 68 294 deaths, with Botswana reporting 24 926 total cases and 202 total deaths [1]. Vaccines are considered one of the most awaited interventions for combating COVID-19 and hundreds of global institutions are working at an unprecedented speed to develop COVID-19 vaccines [2-9]. Several vaccines have been developed and some are still undergoing clinical trials while very few countries have started vaccine rollout [10]. One of the challenges towards COVID-19 vaccination is the uncertainty of vaccine acceptance among the public. In general factors that influence vaccine acceptance include the public’s demand for vaccine, their perception towards the disease and attitudes towards the vaccine [11]

Acceptance and perception studies are important as they provide critical information for health education programs to increase uptake of vaccines and target certain populations. Few studies have been conducted assessing the acceptance, perceptions and attitudes towards COVID-19 vaccines as well as factors influencing uptake which may vary from country to country. A global survey of potential acceptance of a COVID-19 vaccine has shown that differences in acceptance rates ranged from almost 90% (in China) to less than 55% (in Russia) [10].In an Australian study [12], eighty per cent (80%) respondents generally held positive views towards COVID-19 vaccination while 65% of participants in Saudi Arabia [2], showed interest to accept the COVID-19 vaccine if available. The preliminary results of a Chilean case study on COVID-19 vaccine perception in the country showed that 87% were willing to vaccinate, a relatively high proportion and slightly lower than the rate found by García and Cerda [13] which was 90.6% for Chile. Gender and age disparities in attitudes were characterised in Australia, where females (83%) were found to be more likely to depict an optimistic outlook to receiving vaccinations than males (78%), as well as those aged 70 years and above (91%) compared to 76% of 18–29-year-olds [12]. Furthermore, willingness to accept future COVID-19 vaccines in Saudi Arabia was relatively high among older age groups, married participants, participants with a postgraduate degree or higher education level (68.8%), non-Saudi (69.1%), employed in government sector (68.9%) [2]. There is subcultural diversity as African Americans and Hispanics demonstrated higher vaccine hesitancy than other cultural or ethnic groups in a US study [16].

Religion, female gender, residing in deprived neighbourhoods were some factors found to be correlated with COVID-19 vaccine hesitancy in Australia (Edwards et al, 2021), while reliance on social media and refraining from news (Chadwick et al, 2021) were also associated with vaccine hesitancy in the UK.

These studies demonstrate significant variations in COVID-19 vaccine acceptance across different countries and the roles of several socio-demographic determinants of health in vaccine acceptance and risk perception. Such information is critical for context and country specific implementation of COVID-19 vaccination programs, with governments being encouraged to understand communities’ concerns and identify strategies that will support engagement to support effective launching of new vaccines [12]

Circumstantial conditions such as the pandemic context, specifically disease prevalence in a particular population can also impact vaccination intention [13]. Perceived severity of COVID-19 and perceived vaccine safety were the two strongest determinants of vaccine acceptance in a Finland study [14]. In Turkey and the UK, acceptance rates of vaccines were found to be higher among study participants who believed in the natural origin of COVID-19 in contrast to those who believed that the disease was generated by humans [15]. Moreover, credible sources of information about vaccines such as government were reported to instil high levels of trust amongst the public [10]. 

In the African setting, the level of vaccine acceptance (53.6%) and risk perception of (46.7%) were relatively average in Western Uganda [17]. Males, those with a tertiary education, students and non-salary earners were likely to accept the vaccine [17]. 

The Botswana government has adopted a multi-pronged strategy as part of the response to COVID-19. One of the pillars of the COVID-19 epidemic control measures includes vaccinating 276, 078 (16.5%) of the targeted population in the 1st phase of the vaccination campaign. Botswana is a signatory to the WHO/Worldbank vaccine initiative and expects to receive its first COVID-19 vaccines in March 2021. Acceptance and risk perception for COVID-19 vaccines is unknown in Botswana. There have been several conspiracy theories around COVID-19 vaccines which are mostly linked to religious and cultural beliefs which may influence the uptake of the vaccine [19]. The widely used social media platforms have also reported on the negative aspects of the vaccine and its side effects which may also influence the population’s attitudes and perceptions resulting in low acceptance [20]. We therefore assessed the acceptance and risk perceptions of COVID-19 vaccines in Botswana in order to inform the planned population roll out of the vaccines. Demographic, experiential and socio-cultural factors were also explored for their association with the outcome measures.”

8. Study population: you just mentioned that stratified sampling was used to select study participants but offer on more details. How was this done exactly? Were the population size for each of the zones factored into the number recruited per zone? You do not tell us how the 9 zones were created and how that related to the study design.

Response: Thank you for highlighting this to us. We have reviewed our Methods section and used Strobe guidelines to re-write this section so that it is well understood. We have also included a table that shows how this stratified sampling was done and number recruited per each Zone. The new sections reads

“Study Size

The survey leveraged on the already existing resources. The nine (9) COVID-19 zones were used as strata (Table 1). A stratified sampling was employed for this survey. Individuals/study participants were selected within the localities randomly selected from the COVID-19 zones. 

To compute the sample size for the survey a margin of error of 0.05 and a confidence level of 0.95 were used. We further assumed an acceptance rate of 53.6% which is similar to the one estimated by Echoru I et al (2020) in Uganda [19] resulting in a sample of size 383. To cover the 9 COVID-19 zones, we estimated a sample size of 383 multiplied by nine (9) zones giving a total of 3447. To accommodate the design effect, since our sampling strategy is stratified sampling scheme, we assumed a design effect of 1.5 which then gave us a sample size of 5171. Finally, we assumed 90% response rate which resulted in a total sample of 5745 for the survey.

Table 1. The 9 COVID-19 Zones, Sample Size and Allocation

Zone Population Proportional Sample

 Localities >=18 years Locality Individuals

Greater Gaborone 167 590,689 3 2712

Greater Palapye 78 188,428 2 865

Maun 52 84,449 1 388

Greater Francistown 80 192,483 2 884

Chobe 18 23,789 1 109

Ghanzi 17 22,474 1 103

Kgalagadi 47 30,678 1 141

Selebi Phikwe 21 74,678 1 343

Boteti 20 43,810 1 201

Total 500 1,238,768 13 5745

9. Then you say “Individuals/study participants were selected within the localities randomly selected from the Covid-19 zones”, how was this done?

Response: Thank you for this clarification. This comment has also been in addressed in comment number 8 above which details how the stratified sampling was done.

10. Was there any exclusion criteria

Response: Thank you for this highlight. There was inclusion criteria and we added the exclusion criteria in the manuscript and it reads

“The study population consisted of 5300 adults aged 18 years and above since most candidate vaccines were investigated and tried in this group worldwide. In addition, this was in line with the Botswana government policy for COVID-19 vaccine eligibility. Study participants were selected from localities in each of the nine (9) Botswana COVID-19 zones using stratified sampling. Eligibility for participation in the study included those aged 18 years and above including pregnant women, and those with (and without) comorbidities and able to give informed consent. Following consent, participants were enrolled into the study by administering the questionnaires in the selected localities. This happened at government departments, shops, markets and private companies. Exclusion criteria included those who are under 18 years of age and those that did not consent to participate in the study.”

11. You have “To cover the 9 COVID-19 zones, we estimated a sample of 383 multiplied by giving a total of 3447”, there is something missing. It does not read well to me, kindly have a look.

Response: Thank you for this insightful comment. We have noted this mistake and corrected it. The statement now reads as follows

“To cover the 9 COVID-19 zones, we estimated a sample size of 383 multiplied by nine (9) zones giving a total of 3447. To accommodate the design effect, since our sampling strategy is stratified sampling scheme, we assumed a design effect of 1.5 which then gave us a sample size of 5171. Finally, we assumed 90% response rate which resulted in a total sample of 5745 for the survey.”

12. Since this was in-person interviewer administered questionnaire, how was the safety of everyone involved assured?

Response: Thank you for this clarification. We have corrected the statement. This were self-administered questionnaires. 

13. Results: if this was interviewer administered why do you have so many missing values for almost each of the variables?

Response: Thank you for insightful comment. We have noted the error made in the manuscript. This has been corrected in the manuscript, Questionnaires were self-administered.

14. Consider presenting table 2 and 4 as figures rather, I think they will be nicer e.g. as a pie chart or other appropriate figure type. And most importantly, please indicate the 95%CI for the acceptance rates

Response: Thank you for this insightful comment. We have added 95%CI and also converted the tables into pie charts as advised. 

15. You have just too many tables. Apart from those I have pointed out can be figures, there are still a number of tables that can be put together as 1 larger table and will still be clear. Have a look and do this for tables 6-8 and then 9-10.

Response: Thank you for this insightful comments: we have converted some of the tables as advised to figures to reduce the number of tables. We have also merged table 6 and 7 to make one table to reduce the number of table in the manuscript as advised.

16. You mentioned multivariate analysis but I see no such results presented

Response: Thank you for highlighting this to us. We have revised our results section and added more analysis to the tables and figures.

17. Check and deal with some few long sentences and grammatical errors.

Response: thank you for this clarification. We have reviewed the whole document and dealt with the grammatical errors identified in the manuscript.

Reviewer 3 Comments. I have some comments about the manuscript regarding the content of tables, result part, and some minor language issues that follow below. I would strongly recommend that the authors recheck the numbers in the tables.

Results

1) It seems that an error exists in Table 1, page 11 regarding total number of “education level” which is 5310?! Why not 5300?

Response: Thank you for highlighting this to us. Table 1 has been revised and the figures has been corrected.

2) In Table 2, page 12: why total number is not 5300? If missing data exist, I suggest that the authors add them to the table

Response: Thank you for this clarification. We have noted this mistake and this was because of missing data. The table has now been correct to include missing data.

3) Page 12, the part d of the results about “Factors associated with acceptance for the COVID vaccine in Botswana”: as all the factors have been explained, a short explanation about how occupations were associated with acceptance of the COVID vaccine is good to be added to the text

Response: Thank you for this insightful comment. A new and simple Table 3 has been added and the interpretation is also provided.

4) Regarding Table 4, page 13, as total number is 4784, whether 516 is missing here? Then it is better to provide info about missing data in the table

Response: thank you for this clarification. We have noted this error and the number has been fixed and the Table numbers has been revised.

5) Regarding Table 6, page 14, are there missing data? then it is better to provide the number of missing data in the table

Response: Thank you for highlighting this to us. The error was noted and the number has been fixed and the Table numbers has been revised.

6) Regarding Table 7, page 15, I think it is better to add info about missing data if exist

Response: Thank you for highlighting this to us. The number has been fixed and the Table numbers has been revised.

7) In Table 8: it seems that an error exists regarding total numbers: 1172+3772

Response: Thank you for highlighting this to us. The error has been noted and the number has been fixed and the Table numbers has been revised.

8) In Table 9, page 16: it seems that an error exists here. Regarding total frequencies, why is it 5931? and percent which is 111,9 ?!

Response: Thank you for highlighting this to us. The error has been noted and the number has been fixed and the Table numbers has been revised.

9) Table 10: It seems that an error exists here too regarding total frequencies: why is it 6203? and percent which is 117 ?!

Response: Thank you for highlighting this to us. The mistake has been noted and the number has been fixed and the Table numbers has been revised.

*And some minor language issues:

Page 15 line 2: Forty nine percent of participants who said their religion and “cultural” hinders vaccine uptake were willing to receive the COVID-19 vaccine. use the word "culture" instead of cultural

Response: thank you for the insightful comment. We have done as advised and the new statement reads as follows “Willingness to take the Vaccine against religion and culture”

Regarding title of Table 7, delete the first “believes” and add "religious" instead of religion

Response: thank you for this insightful comment: We have noted this and have corrected it to read as follows” Participants whose religious beliefs did not hinder vaccination were more likely to take the vaccine than those whose religious beliefs hinder vaccination. Table 47 shows that almost about half (49%) of the participants who said their religious and cultural beliefs hinder vaccine uptake were willing to receive the COVID-19 vaccine”

Page 15: regarding the line that follows “Willing to receive COVID vaccine though trust other traditional and religious methods over vaccine” delete i

Response: Thank you for this comment. We have corrected the mistake

Reviewer 4 comments: Thank you for considering me to review this manuscript, “Acceptance Rate and Risk Perception towards the COVID-19 Vaccine in Botswana: Across-Sectional Study”. As the only viable option to mitigate the COVID-19 pandemic is via vaccination, this implies understanding vaccination hesitancy is paramount so that factors that contribute to vaccination hesitancy could be recognized so that effectively counteract measures -health education - could be contemplated. This study purports to explore the acceptance rate and risk perception of COVID-19 vaccines amongst the general population in southern Africa country, Botswana. The authors developed their questions to solicit acceptance rate and risk perception. The authors also included various socio-demographic information and clinical risk factors. The result seems interesting but the authors need to address some issues to render their manuscript with high calibre scientific merit.

Response: Thank you for the insights. We will work on the feedback and respond appropriately.

Comment •TITLE: The content of the study -the instrument used- is currently not clear to the readers. I would think here that the title would once the outcome measures are made explicit in the method section.

Response: Thank you for highlighting this to us. We have worked on our title and instrument used by responding appropriately to the comments made on the method section.

Comment •ABSTRACT: The ABSTRACT in general has been structured according to the style of this journal. Minor issues: change “Introduction” to “Background”. During COVID, there is a ‘tradition’ in most journals to include the dates of the data collection (1-28 February 2021).

It seems there is the repetition of the aims (“This study, aims to assess the acceptance rate and risk perception of COVID-19 vaccines amongst the general population in Botswana” … “This study aims to assess the acceptance rate and risk perception of COVID-19 vaccines amongst the general population in Botswana”. SUGGESTION, keep one and create a subheading -Method. In the method, bring outcomes measures on board and the fact the study explored socio-demographic information and some risk factors.

This information is not needed (…” and 31.3% of participants”) (“At P<0.05, the following...”. Suggestion: delete

Word employed should be geared toward the outcome measures of the study.

KEYWORD. This keyword should be included (“vaccine hesitancy”)

Response: The comments are all welcome and have been incorporate into the manuscript text. The abstract now reads as follows:

“Background: The COVID-19 disease burden continues to be high worldwide and vaccines continue to be developed to help combat the pandemic. Acceptance and risk perception for COVID-19 vaccines is unknown in Botswana despite the government’s decision to roll out the vaccine nationally. 

Objectives: This study aims to assess the acceptance rate and risk perception of COVID-19 vaccines amongst the general population in Botswana. 

Methods: We interviewed 5300 adults in Botswana from 1-28 February 2021 using self-administered questionnaires. The main outcomes of the study were vaccine acceptance and hesitancy rates. Demographic, experiential and socio-cultural factors were explored for their association with outcome variables.

Results: Two-thirds of the participants were females (3199), with those aged 24-54 making the highest proportion (61%). The acceptance rate of COVID-19 vaccine was 73.4% with vaccine hesitancy at 31.3%. Factors found to be associated with willingness to take the COVID-19 vaccine were sex (p= 0.033), age group (p= 0.009), education level (p= 0.000), willingness to wear face mask (p=0.000), religious beliefs (p=0.000), occupation (p=0.023) and marital status (p=0.045). Safety perception of the vaccine was associated with education level (p= 0.000), willingness to wear face mask (p=0.000), age group (p=0.000) and religion (p=0.001). Ninety six percent (96%) of those willing to receive the vaccine indicated willingness to continue wearing masks and social distancing after vaccination. Radio and Television were the preferred sources for COVID-19 prevention and vaccination information. 

Conclusion: This study demonstrated a high acceptance rate for the COVID-19 vaccine and a low risk perception in Botswana. In order to achieve a high vaccine coverage and ensure a successful vaccination process, there is need to target populations with high vaccine hesitancy rates. A qualitative study to assess the factors associated with vaccine acceptance and hesitancy is recommended to provide an in-depth analysis of the findings.”

 KEYWORDS: Vaccine hesitancy added to the key words. Keywords now read;

“Acceptance Rate, vaccine hesitancy, Risk Perception COVID-19 Vaccine, Case, Contact and Contact Tracing””

Comment: •BACKGROUND INFORMATION (INTRODUCTION)

The background information appears to lead the reader to sense what is coming up as the aims of this study -well done.

This statement should be rephrased (“Vaccine acceptance mirrors the public’s perception towards the disease threat, demand for and attitude towards the vaccine”)

The word ‘race’ (“With regards to race, African-Americans and Hispanics demonstrated higher vaccine hesitancy than other races in a US study [16].”) is confusing. Suggestion: ethic group or simple state as following (“There is subcultural diversity, African-Americans and Hispanics demonstrated higher vaccine hesitancy than other cultural or ethnic groups in a US study”.

To avoid too much rambling (“Global literature has also recognized the distinctive roles of several socio-demographic determinants of health in vaccine acceptance and risk perception. Additionally, ….Moreover, credible sources of information about vaccines such as government were reported to instil high levels of trust amongst the public[10].”). Since these paragraphs are preceded by a literature view on the rate of vaccine hesitancy, this paragraph should simply articulate ‘associated factors of vaccine hesitancy’. For a better flow of information, the narration on the rate of vaccine hesitancy should be sent to the previous paragraph.

For aims (“We therefore assessed the acceptance and perceptions of COVID-19 vaccines in Botswana in order to inform the planned population roll out of the vaccines.”), it should reflect the content of the instrument used or developed. Also, the associated factors explored should be mentioned as one of the aims.

Response: Thank you for this suggestion. We have amended the background as suggested. It now reads:

“ Background

According to the World Health Organisation (WHO) weekly epi update of the 14th February 2021, there were a total of 108 246 992 COVID-19 cases and 2 386 717 COVID-19 deaths [1]. Of these, the African region reported a total of 2 723 431 COVID-19 cases and 68 294 deaths, with Botswana reporting 24 926 total cases and 202 total deaths [1]. Vaccines are considered one of the most awaited interventions for combating COVID-19 and hundreds of global institutions are working at an unprecedented speed to develop COVID-19 vaccines [2-9]. Several vaccines have been developed and some are still undergoing clinical trials while very few countries have started vaccine rollout [10]. One of the challenges towards COVID-19 vaccination is the uncertainty of vaccine acceptance among the public. In general factors that influence vaccine acceptance include the public’s demand for vaccine, their perception towards the disease and attitudes towards the vaccine [11]

Acceptance and perception studies are important as they provide critical information for health education programs to increase uptake of vaccines and target certain populations. Few studies have been conducted assessing the acceptance, perceptions and attitudes towards COVID-19 vaccines as well as factors influencing uptake which may vary from country to country. A global survey of potential acceptance of a COVID-19 vaccine has shown that differences in acceptance rates ranged from almost 90% (in China) to less than 55% (in Russia) [10].In an Australian study [12], eighty per cent (80%) respondents generally held positive views towards COVID-19 vaccination while 65% of participants in Saudi Arabia [2], showed interest to accept the COVID-19 vaccine if available. The preliminary results of a Chilean case study on COVID-19 vaccine perception in the country showed that 87% were willing to vaccinate, a relatively high proportion and slightly lower than the rate found by García and Cerda [13] which was 90.6% for Chile. Gender and age disparities in attitudes were characterised in Australia, where females (83%) were found to be more likely to depict an optimistic outlook to receiving vaccinations than males (78%), as well as those aged 70 years and above (91%) compared to 76% of 18–29-year-olds [12]. Furthermore, willingness to accept future COVID-19 vaccines in Saudi Arabia was relatively high among older age groups, married participants, participants with a postgraduate degree or higher education level (68.8%), non-Saudi (69.1%), employed in government sector (68.9%) [2]. There is subcultural diversity as African Americans and Hispanics demonstrated higher vaccine hesitancy than other cultural or ethnic groups in a US study [16].

Religion, female gender, residing in deprived neighbourhoods were some factors found to be correlated with COVID-19 vaccine hesitancy in Australia (Edwards et al, 2021), while reliance on social media and refraining from news (Chadwick et al, 2021) were also associated with vaccine hesitancy in the UK.

These studies demonstrate significant variations in COVID-19 vaccine acceptance across different countries and the roles of several socio-demographic determinants of health in vaccine acceptance and risk perception. Such information is critical for context and country specific implementation of COVID-19 vaccination programs, with governments being encouraged to understand communities’ concerns and identify strategies that will support engagement to support effective launching of new vaccines [12]

Circumstantial conditions such as the pandemic context, specifically disease prevalence in a particular population can also impact vaccination intention [13]. Perceived severity of COVID-19 and perceived vaccine safety were the two strongest determinants of vaccine acceptance in a Finland study [14]. In Turkey and the UK, acceptance rates of vaccines were found to be higher among study participants who believed in the natural origin of COVID-19 in contrast to those who believed that the disease was generated by humans [15]. Moreover, credible sources of information about vaccines such as government were reported to instil high levels of trust amongst the public [10]. 

In the African setting, the level of vaccine acceptance (53.6%) and risk perception of (46.7%) were relatively average in Western Uganda [17]. Males, those with a tertiary education, students and non-salary earners were likely to accept the vaccine [17]. 

The Botswana government has adopted a multi-pronged strategy as part of the response to COVID-19. One of the pillars of the COVID-19 epidemic control measures includes vaccinating 276, 078 (16.5%) of the targeted population in the 1st phase of the vaccination campaign. Botswana is a signatory to the WHO/Worldbank vaccine initiative and expects to receive its first COVID-19 vaccines in March 2021. Acceptance and risk perception for COVID-19 vaccines is unknown in Botswana. There have been several conspiracy theories around COVID-19 vaccines which are mostly linked to religious and cultural beliefs which may influence the uptake of the vaccine [19]. The widely used social media platforms have also reported on the negative aspects of the vaccine and its side effects which may also influence the population’s attitudes and perceptions resulting in low acceptance [20]. We therefore assessed the acceptance and risk perceptions of COVID-19 vaccines in Botswana in order to inform the planned population roll out of the vaccines. Demographic, experiential and socio-cultural factors were also explored for their association with the outcome measures.”

Comment •MATERIALS AND METHODS: The problem with the METHOD is that the authors failed to elaborate on the development of the questionnaire used. Further, there is no information on the reliability and other important information on its applicability to the present population. Please attend to this issue.

Overall, the method has much to be desired. I would encourage the authors to stick with the subheadings that are recommended by Strobe observational study checklist (https://www.strobe-statement.org/index.php?id=available-checklists).

This statement touches on an important part of the study (“Following consent, participants were enrolled into the study by administering the questionnaires in the selected localities. This happened at government departments, shops, markets and private companies”.). How data collection was done should be more explicit in the text. Please identify the name of the technique for data collection (probabilistic sampling as stated in the limitation? ).

Different parts of the text describing recruiting and data collection should be consistent with each other. Thus, strict adherence to Strobe observational study checklist would help.

For us international reader, we may not understand what constitute “across the nine (9) COVID-19 zones in Botswana”. Please define.

The rationale for including this statement (“A risk perception model which integrates three core dimensions; cognitive factors (knowledge), experiential factors (emotion) and socio-cultural factors (norms, values) was used to guide in the development of the questions”) is needed. Also, a separate section is needed to highlight the development of the questionnaire. This should be merged with this (“designed from literature review [12, 21, 22]”).

Any reference number was assigned for the present ethical approval from IRB?

The result section has a lot of information and some of them were not described in the methodology or stated as the aim of the study. This is major misgiving of this manuscript. The confusion partly stems from the factor that outcome measures were not described in the method.

Information in different tables should be merged into coherent themes.

Response: Thank you for clarifying this to us. We reviewed our methodology section and used the strobe guidelines to write our methods section clearly. The new method section is as below.

“Materials and Methods

Study Design

This study used a cross sectional survey design conducted across the nine (9) COVID-19 zones in Botswana from 1-28 February 2021 using enumerator administered questionnaires designed from literature review and using a risk perception model which integrates three core dimensions; cognitive factors (knowledge), experiential factors (emotion) and socio-cultural factors (norms, values) [12, 23, 24]. The questionnaires were self-administered without any interference from the interviewer. 

Study Setting 

Botswana is a landlocked country situated in Southern Africa and shares borders with South Africa, Namibia, Zimbabwe and Zambia. The dynamics of population movement across borders with neighbouring countries increases the risk of transmission of communicable diseases including COVID-19. Botswana has an estimated population of 2.374,697, (1,171,629 Males and 1,203,068 Females) as per the 2011 population census projection for 2020 [22]. The population is spread over vast land of 581,730km2. The estimated population density in Botswana is 4 per Km. Sixty nine percent (69.4%) of the total population lives in urban settings.

Botswana’s economy is largely dependent on mineral revenues and belongs to Southern Africa Custom Union. The Gross National Income (GNI) per capita is approximately $6,000, thus by classification, Botswana belongs to the upper middle-income countries. Botswana recorded its first COVID-19 case in March 2020. After recording the first case the country was demarcated into nine (9) sections called COVID-19 zones in order to restrict movement of non-essential travel between these zones. In order to travel between these zones, a valid permit, which should be applied for online, is necessary. 

Participants 

The study population consisted of 5300 adults aged 18 years and above since most candidate vaccines were investigated and tried in this group worldwide. In addition, this was in line with the Botswana government policy for COVID-19 vaccine eligibility. Study participants were selected from localities in each of the nine (9) Botswana COVID-19 zones using stratified sampling. Eligibility for participation in the study included those aged 18 years and above including pregnant women, and those with (and without) comorbidities and able to give informed consent. Following consent, participants were enrolled into the study by administering the questionnaires in the selected localities. This happened at government departments, shops, markets and private companies. Exclusion criteria included those who are under 18 years of age and those that did not consent to participate in the study.

Study Size

The survey leveraged on the already existing resources. The nine (9) COVID-19 zones were used as strata (Table 1). A stratified sampling was employed for this survey. Individuals/study participants were selected within the localities randomly selected from the COVID-19 zones. 

To compute the sample size for the survey a margin of error of 0.05 and a confidence level of 0.95 were used. We further assumed an acceptance rate of 53.6% which is similar to the one estimated by Echoru I et al (2020) in Uganda [19] resulting in a sample of size 383. To cover the 9 COVID-19 zones, we estimated a sample size of 383 multiplied by nine (9) zones giving a total of 3447. To accommodate the design effect, since our sampling strategy is stratified sampling scheme, we assumed a design effect of 1.5 which then gave us a sample size of 5171. Finally, we assumed 90% response rate which resulted in a total sample of 5745 for the survey. Table 1. The 9 COVID-19 Zones, Sample Size and Allocation

Zone Population Proportional Sample

 Localities >=18 years Locality Individuals

Greater Gaborone 167 590,689 3 2712

Greater Palapye 78 188,428 2 865

Maun 52 84,449 1 388

Greater Francistown 80 192,483 2 884

Chobe 18 23,789 1 109

Ghanzi 17 22,474 1 103

Kgalagadi 47 30,678 1 141

Selebi Phikwe 21 74,678 1 343

Boteti 20 43,810 1 201

Total 500 1,238,768 13 5745

Variables

The exposure variables for this study included age, gender, level of education, Occupation, religious background, Media, marital status and place of residence. The primary outcome measure includes acceptance rate and risk perception towards the COVID-19 vaccine.

Bias

Participation was ere selected on a voluntary basis therefore the survey had potential for self-selection bias by community members who are particularly concerned about the pandemic. However, probabilistic sampling was employed. The study also used self-administered questionnaires and this could have an impact on the response rate by the participants. 

Data Sources

The cross-sectional survey used questionnaires to interview participants from selected areas across the country. The questionnaires were written in both English and Setswana and data obtained from the questionnaire were analysed using the STATA statistical software version 15. The questionnaires were validated before being administered to all the participants. The questionnaires were The participants answered the same questionnaire across zones to obtain reliable information. Data protection was ensured via file password protection and limiting access of information to the project team members only. 

Statistical Methods

Categorical variables were analysed using summary statistics such as frequencies and percentages. For continuous variables such as age, median (inter-quartile range) or mean (and standard deviation) were presented depending on whether the variable under consideration is skewed or not, respectively. Acceptance was measured by the question asking respondents if they are willing to receive the COVID-19 vaccine when it is rolled out nationally. The acceptance rate of COVID-19 vaccine was the number that “accept” divided by the sample size but excluding those with missing responses. The independent variables were demographic variables such as age, gender, marital status, educational level, and other important variables such presence and absence of comorbidities, COVID-19 zone level (high/red zone, medium, low), socio-cultural and experimental factors. Risk perception was measured by the questions asking respondents if they considered the COVID-19 vaccine to be safe.

Bivariate analysis using the chi-square test was conducted to determine factors associated with both acceptance and risk perception for the COVID-19 vaccine. Multiple logistic regression was conducted to adjust for any confounding factors. Odds ratios, as measures of effect, and the corresponding p-values and 95% confidence intervals are presented. A p-value less than 0.05 was used to determine significance.

Ethical considerations

This study was conducted according to Botswana, and International Standards of Good Clinical Practice, applicable government regulations and Institutional research policies and procedures. The protocol and any amendments made were submitted to the Botswana, Ministry of Health and Wellness Institutional Review Board (IRB), were it was approved. The protocol number awarded for this study is HPDME 13/18/1 

All subjects were asked to provide written informed consent before responding to questions and they were free to withdraw at any time during the study.”

•DISCUSSION I am not going to dwell too much on the discussion section since would invariably be changed if the above-mentioned suggestions are contemplated

Response: Thank you for highlighting this to us we have worked on our discussion section to be aligned to the methods and objectives of the study. 

•ACKNOWLEDGEMENT 

You want to separate acknowledged name (ACKNOWLEDGMENT) person by coma rather than bullets

Response: Thank you for highlighting this to us. We have corrected this mistake in the Acknowledgement section

•REFERENCES

The authors have employed 30 references. Most of them are relevant and update.

Response: Thank you for this comment. We will continue to use relevant and updated literature.

All authors have seen and approved the manuscript, and have contributed to the work. There are no copyright restrictions. No authors have conflicts of interest to disclose.

Thank you for considering our revised manuscript. Please contact me for additional information.

Sincerely,

Lebapotswe B Tlale

Public Health Specialist

Ministry of health and Wellness

Cell: +267 71416607 and +267 73340732

Email: cypro330@yahoo.com

---

## [Decision Letter · Decision Letter 1]

9 Sep 2021

PONE-D-21-09673R1Acceptance Rate and Risk Perception towards the COVID-19 Vaccine in BotswanaPLOS ONE

Dear Dr. Tlale,

Thank you for submitting your manuscript to PLOS ONE. After careful consideration, we feel that it has merit but does not fully meet PLOS ONE’s publication criteria as it currently stands. Therefore, we invite you to submit a revised version of the manuscript that addresses the points raised during the review process.

The reviewers have again raised some important issues which we believe should be addressed in order to strengthen the manuscript for publication.

We look forward to receiving your revised manuscript.

Kind regards,

David Teye Doku

Academic Editor

PLOS ONE

Journal Requirements:

Reviewers' comments:

Reviewer's Responses to Questions

**Comments to the Author**

1. If the authors have adequately addressed your comments raised in a previous round of review and you feel that this manuscript is now acceptable for publication, you may indicate that here to bypass the “Comments to the Author” section, enter your conflict of interest statement in the “Confidential to Editor” section, and submit your "Accept" recommendation.

Reviewer #1: (No Response)

Reviewer #2: (No Response)

2. Is the manuscript technically sound, and do the data support the conclusions?

Reviewer #1: Yes

Reviewer #2: Yes

3. Has the statistical analysis been performed appropriately and rigorously? 

Reviewer #1: Yes

Reviewer #2: Yes

4. Have the authors made all data underlying the findings in their manuscript fully available?

Reviewer #1: Yes

Reviewer #2: Yes

5. Is the manuscript presented in an intelligible fashion and written in standard English?

Reviewer #1: Yes

Reviewer #2: Yes

6. Review Comments to the Author

Reviewer #1: Thank you for considering me to re-review this manuscript and now entitled, “Acceptance Rate and Risk Perception towards the COVID-19 Vaccine in Botswana. This manuscript was scrutinized by 3 reviewers including myself. The authors have addressed all the comments raised by the reviewers. Both the conceptual and scientific merit of the manuscript has now improved Some minor issues.

Under “Qualifications and emails of all authors”, some of the authors were not accompanied their qualifications. Any reason?

In the method, maybe this statement (“Bias 203 Participation were selected on a …. rate by the participants) could be considered as a limitation. This could be transferred to the Discussion. However, the authors need to expand on the employed probabilistic sampling

The strength of this study is the fact that the authors developed the study questionnaire and it dispensed in both languages (English and Setswana). It would be nice if the authors insert one paragraph or so on the development and validation of the questionnaire. How many items were on the questionnaire? Maybe identify the themes covered (e.g., Demographic information, Cognitive Factors, Experiential Factors, and Socio-cultural factors). The authors could give examples or rationale for each factor.

One item of the questionnaire (Section B – Cognitive Factors) B1 asks, “Are you suffering from any of the following ….). I am not sure whether this cognitive factor or is it?

I would beg to differ here from other reviewers by recommending the deletion of Figure 1 and Figure 1. First of all, this manuscript is already a bulk one. Also, these figures do not add anything new. This information could be simply narrated in the text.

Overall, this is important and well conducted work and I would advocate publication if those minor issues can be properly addressed.

Reviewer #2: This is surely an improved version of the manuscript. These few things I believe if addressed will make it even better:

1. Generally your table titles need to be improved such that the tables can stand alone and still be readily meaningful to the reader e.g. Tables 1 and 3 could end with (N= 5300). Table 4 title is difficult to understand so rephrase it to be complete and clearly understood. Most titles currently end with “in Botswana” but this was just among the study participants so commonly that is what the title should indicate, something like ‘among study participants in Botswana’

2. Even though you agreed to attach the questionnaire as supplementary file, there is still the need in the methods section to briefly describe the development of the questionnaire, the content and any piloting processes etc.

3. You have modified to say, this was self-administered questionnaire, does that imply that everyone who participated could read and write in English or the local language used? That would mean that anyone who could not read and write the 2 languages was excluded? You still need to clarify this process.

4. Tables 5 and 6 can easily be one table

5. Now you have multivariate analysis results so I would expect that to influence what is presented in the abstract results section but I don’t seem to see that.

7. PLOS authors have the option to publish the peer review history of their article (what does this mean?). If published, this will include your full peer review and any attached files.

Reviewer #1: **Yes: **Samir Al-Adawi

Reviewer #2: No

---

## [Author Response · Author response to Decision Letter 1]

16 Sep 2021

Dr Lebapotswe Bahumi Tlale, MBBS

 15th September 2021

David Teye Doku

Academic Editor

PLOS ONE

Dear Sir

RE: Manuscript ID PONE-D-21-09673 Acceptance Rate and Risk Perception towards the COVID-19 Vaccine in Botswana: A Cross-Sectional Study. PLOS ONE

Thank you for your and the reviewer’s thoughtful review of our manuscript. With these in mind we are pleased to submit a revised manuscript for publication in the PLOS ONE Journal. The manuscript has not been published, and is not being considered for publication elsewhere.

These are our responses to each point raised by the academic editor and the reviewers:

Reviewer #1: 

Comment 1: Thank you for considering me to re-review this manuscript and now entitled, “Acceptance Rate and Risk Perception towards the COVID-19 Vaccine in Botswana. This manuscript was scrutinized by 3 reviewers including myself. The authors have addressed all the comments raised by the reviewers. Both the conceptual and scientific merit of the manuscript has now improved some minor issues.

Response: Thank you for reviewing our manuscript and highlighting the minor issues to be addressed

Comment 2: Under “Qualifications and emails of all authors”, some of the authors were not accompanied their qualifications. Any reason?

Response: Thank you for highlighting this to us. We have edited this section to include all the authors’ qualification. The revised version is as follows:

“Qualifications and emails of all authors:

Lebapotswe B. Tlale, MBBS (UWI), MMed Public Health UB) cypro330@yahoo.com

Lesego Gabaitiri BA, MSc, PhD gabaitiril@biust.ac.bw

Lorato Totolo Diploma, BA community health, PGD kaone.regoeng@gmail.com

Gomolemo Smith, MBBS (UB) gomolemodisang@yahoo.com

Orapeleng Puswane-Katse, MBBS (UWI) MMed (UB) ophuswane@yahoo.co.uk

Eunice Ramonna, BA social work (UB), MPH (Pretoria) lapoeunx@gmail.com

Basego Mothowaeng, BA Communication Science basego07@gmail.com

John Tlhakanelo, MBCHB, MMed (UB) tlhakaneloj@ub.ac.bw

Tiny Masupe MBCHB, MSC, MPH tiny.masupe@mopipi.ub.bw

Goabaone Rankgoane-Pono, MBCHB (UPT), MMeD (UB), Dip HE (UCT)goaba2000@yahoo.com

John Irige, MPH jirige@fhi360.org

Faith Mafa, BA media production faithmorwaagole@gmail.com

Samuel Kolane, BPH Health Education sinki.kolane@gmail.com”

Comment 3: In the method, maybe this statement (“Bias 203 Participation were selected on a …. rate by the participants) could be considered as a limitation. This could be transferred to the Discussion. However, the authors need to expand on the employed probabilistic sampling

Response: thank you for highlighting this to us. We had in the previous version of this manuscript included this paragraph under study limitation in the discussion section but was advised by one of the reviewers to move it to the methodology section. We have now moved it back to the discussion section under study limitation and it now reads.

“Participation was on a voluntary basis therefore the survey had potential for self-selection bias by community members who are particularly concerned about the pandemic. However, probabilistic sampling (stratified Sampling method) was employed. The study also used self-administered questionnaires and the disadvantages of self-administered questionnaires is low response rates, exclusion of those who cannot read and write, and that the researcher cannot couch for the validity of the responses from self-administered surveys. However in our study the response rate was high more than 95 % and also around 95.7% of participants had at least attended primary school level.”

Comment 4: The strength of this study is the fact that the authors developed the study questionnaire and it dispensed in both languages (English and Setswana). It would be nice if the authors insert one paragraph or so on the development and validation of the questionnaire. How many items were on the questionnaire? Maybe identify the themes covered (e.g., Demographic information, Cognitive Factors, Experiential Factors, and Socio-cultural factors). The authors could give examples or rationale for each factor.

Response: Thank you for highlighting this to us. We have include a paragraph as advised by the reviewer and the paragraph reads:

“To ascertain quality, the questionnaire was pretested before the final draft was made. The development of the questions was guided by the WHO Technical Advisory Group on Behavioural Insights and Sciences for Health paper entitled “Behavioural Considerations for Acceptance and Uptake of COVID-19 Vaccines”. The draft contained about 7 themes and some were dropped because some of the questions were similar or a repetition of questions in other themes. The final version contained Demographic information, Cognitive Factors, Experiential Factors, and Socio-cultural factors. The final version was translated into the native official language and back into English language.”

Comment 5: One item of the questionnaire (Section B – Cognitive Factors) B1 asks, “Are you suffering from any of the following ….). I am not sure whether this cognitive factor or is it?

Response: Thank you for this clarification. The question is not part of cognitive factors. It is an error that in the questionnaire it was under cognitive factors. The question asks about Comorbid factors of the participants. During the analysis this error was noted and it was not analysed or interpreted as a cognitive factor.

Comment 6: I would beg to differ here from other reviewers by recommending the deletion of Figure 1 and Figure 1. First of all, this manuscript is already a bulk one. Also, these figures do not add anything new. This information could be simply narrated in the text.

Response: Thank you for this clarification. I have removed figure 1 from the manuscript and replaced it with a text (Narration of the results) as advised by the reviewer. Note that we had in our original manuscript narrated this section and was advised by other reviewers to insert a figure. The edited section now reads

“Three thousand, six hundred and eighty nine (3689) out of five thousand and twenty seven (5027) Participants Figure 1shows that out of 5 027 participants, 3689 were willing to take the vaccine resulting in the acceptance rate of the COVID-19 vaccine in Botswana of 73.4% (CI: 72.2%, 74.6%)”

Overall, this is important and well conducted work and I would advocate publication if those minor issues can be properly addressed.

Reviewer #2: 

This is surely an improved version of the manuscript. These few things I believe if addressed will make it even better:

Comment 1. Generally your table titles need to be improved such that the tables can stand alone and still be readily meaningful to the reader e.g. Tables 1 and 3 could end with (N= 5300). Table 4 title is difficult to understand so rephrase it to be complete and clearly understood. Most titles currently end with “in Botswana” but this was just among the study participants so commonly that is what the title should indicate, something like ‘among study participants in Botswana’

Response: Thank you for this clarification. We have corrected all the tables as advised by the reviewer. The new table titles are as follows

“

• Table 2: Socio-Demographic characteristics of participants (n=5300)

• Table 3: Factors associated with COVID-19 vaccine acceptance rate and safety perception among participants in Botswana. (n=5300)

• Table 4: Willingness to receive COVID-19 vaccine against cultural and religious believes among participants in Botswana

• Table 5: The preferred and trusted source of information dissemination among participants in Botswana

• Table 67: Vaccine acceptance rate and safety ofby COVID-19 vaccine by districts in Botswana.”

Comment 2. Even though you agreed to attach the questionnaire as supplementary file, there is still the need in the methods section to briefly describe the development of the questionnaire, the content and any piloting processes etc.

Response: Thank you for highlighting this to us. We have include a paragraph as advised by the reviewer and the paragraph reads:

“To ascertain quality, the questionnaire was pretested before the final draft was made. The development of the questions was guided by the WHO Technical Advisory Group on Behavioural Insights and Sciences for Health paper entitled “Behavioural Considerations for Acceptance and Uptake of COVID-19 Vaccines”. The draft contained about 7 themes and some were dropped because some of the questions were similar or a repetition of questions in other themes. The final version contained Demographic information, Cognitive Factors, Experiential Factors, and Socio-cultural factors. The final version was translated into the native official language and back into English language.”

3. You have modified to say, this was self-administered questionnaire, does that imply that everyone who participated could read and write in English or the local language used? That would mean that anyone who could not read and write the 2 languages was excluded? You still need to clarify this process.

Response: Thank you highlighting this to us. We have included this in our limitation to read 

“The study also used self-administered questionnaires and the disadvantages of self-administered questionnaires is low response rates, exclusion of those who cannot read and write and that the researcher cannot couch for the validity of the responses from self-administered surveys. However in our study the response rate was high more than 95 % and also around 95.7% of participants had at least attended primary school level”

Comment 4. Tables 5 and 6 can easily be one table

Response: Thank you for highlighting this to us. We have combined table 5 and 6 and also the narratives of the 2 tables. The new edited section reads as follows:

e. “Preferred and trusted mode of COVID-19 vaccine related communication

Table 5 shows that radio ranked as the number 1 (50.3%) preferred source of information followed by television. The table also shows that the majority (53.3%) of the participants trusted the government most as the source of information for COVID-19, followed by World Health Organization (WHO).

Table 5: The preferred and trusted source of information dissemination among participants in Botswana 

 Preferred source of information dissemination Most trusted source of information

Source of information

 Frequency (%) Rank Source of Information Frequency Rank

 Radio 2664 (50.3) 1 Government 2826 (53.3) 1

 Television 1412 (26.6) 2 WHO 1892 (35.7) 2

 Social media 932 (17.6) 3 Social media 1050 (19.8) 3

 Newspaper 470 (8.9) 4 Internet 435 (8.2) 4

Internet 453 (8.5) 5 

“

5. Now you have multivariate analysis results so I would expect that to influence what is presented in the abstract results section but I don’t seem to see that.

Response: Thank you for this clarification. We have re-edited our abstract and the new edited abstract reads:

“Background: The COVID-19 disease burden continues to be high worldwide and vaccines continue to be developed to help combat the pandemic. Acceptance and risk perception for COVID-19 vaccines is unknown in Botswana despite the government’s decision to roll out the vaccine nationally. 

Objectives: This study aims to assess the acceptance rate and risk perception of COVID-19 vaccines amongst the general population in Botswana. 

Methods: We interviewed 5300 adults in Botswana from 1-28 February 2021 using self-administered questionnaires. The main outcomes of the study were vaccine acceptance and hesitancy rates. Demographic, experiential and socio-cultural factors were explored for their association with outcome variables.

Results: Two-thirds of the participants were females (3199), with those aged 24-54 making the highest proportion (61%). The acceptance rate of COVID-19 vaccine was 73.4% (95% CI: 72.2%-74.6%) with vaccine hesitancy at 31.3% (95% CI: 30.0%-32.6%). When the dependent variable was vaccine acceptance, males had higher odds of accepting the vaccine compared to females (OR=1.2, 95% CI: 1.0, 1.4). Individuals aged 55-64 had high odds of accepting the vaccine compared to those aged 65 and above (OR=1.2, 95% CI: 0.6, 2.5). The odds of accepting the vaccine for someone with primary school education were about 2.5 times that of an individual with post graduate level of education. Finally, individuals with comorbidities had higher odds (OR=1.2, 95% CI: 1.0, 1.5) of accepting the vaccine compared to those without any underlying conditions. Conclusion: This study demonstrated a high acceptance rate for the COVID-19 vaccine and a low risk perception in Botswana. In order to achieve a high vaccine coverage and ensure a successful vaccination process, there is need to target populations with high vaccine hesitancy rates. A qualitative study to assess the factors associated with vaccine acceptance and hesitancy is recommended to provide an in-depth analysis of the findings.

Word Count: 299 (Target 300).”

All authors have seen and approved the manuscript, and have contributed to the work. There are no copyright restrictions. No authors have conflicts of interest to disclose.

Thank you for considering our revised manuscript. Please contact me for additional information.

Sincerely,

Lebapotswe B Tlale

Public Health Specialist

Ministry of health and Wellness

Cell: +267 71416607 and +267 73340732

Email: cypro330@yahoo.com

---

## [Editor Report · Decision Letter 2]

28 Sep 2021

PONE-D-21-09673R2Acceptance Rate and Risk Perception towards the COVID-19 Vaccine in BotswanaPLOS ONE

Dear Dr. Tlale,

Thank you for submitting your manuscript to PLOS ONE. After careful consideration, we feel that it has merit but does not fully meet PLOS ONE’s publication criteria as it currently stands. Therefore, we invite you to submit a revised version of the manuscript that addresses the points raised during the review process.

In your response to Reviwer 2, you indicated that 95.7% of your respondents at least attended primary school. Could you explain how the 4.3% of the respondents who did not attend at least primary school responded to your self-administered questionnaire of that nature. 

Please clarify this before final decision is reached on your manuscript.

“The study also used self-administered questionnaires and the disadvantages of self-administered questionnaires is low response rates, exclusion of those who cannot read and write and that the researcher cannot couch for the validity of the responses from self-administered surveys. However in our study the response rate was high more than 95 % and also around 95.7% of participants had at least attended primary school level” 

We look forward to receiving your revised manuscript.

Kind regards,

David Teye Doku

Academic Editor

PLOS ONE

Journal Requirements:

Additional Editor Comments (if provided):

In your response to Reviwer 2, you indicated that 95.7% of your respondents at least attended primary school. Could you explain how the 4.3% of the respondents who did not attend at least primary school responded to your self-administered questionnaire of that nature.

Please clarify this.

“The study also used self-administered questionnaires and the disadvantages of self-administered questionnaires is low response rates, exclusion of those who cannot read and write and that the researcher cannot couch for the validity of the responses from self-administered surveys. However in our study the response rate was high more than 95 % and also around 95.7% of participants had at least attended primary school level”
---

## [Author Response · Author response to Decision Letter 2]

4 Oct 2021

Dr Lebapotswe Bahumi Tlale, MBBS

 4th October 2021

David Teye Doku

Academic Editor

PLOS ONE

Dear Sir

RE: Manuscript ID PONE-D-21-09673 Acceptance Rate and Risk Perception towards the COVID-19 Vaccine in Botswana: A Cross-Sectional Study. PLOS ONE

Thank you for your and the reviewer’s thoughtful review of our manuscript. With these in mind we are pleased to submit a revised manuscript for publication in the PLOS ONE Journal. The manuscript has not been published, and is not being considered for publication elsewhere.

These are our responses to each point raised by the academic editor and the reviewers:

Editor: 

Comment 1: In your response to Reviwer 2, you indicated that 95.7% of your respondents at least attended primary school. Could you explain how the 4.3% of the respondents who did not attend at least primary school responded to your self-administered questionnaire of that nature. 

Please clarify this before final decision is reached on your manuscript.

“The study also used self-administered questionnaires and the disadvantages of self-administered questionnaires is low response rates, exclusion of those who cannot read and write and that the researcher cannot couch for the validity of the responses from self-administered surveys. However in our study the response rate was high more than 95 % and also around 95.7% of participants had at least attended primary school level”

Response: Thank you highlighting this to us. We have reviewed this section and added the how the 4.3% percent responded and is the new write up is as follows:

“The study also used self-administered questionnaires and the disadvantages of self-administered questionnaires is low response rates, exclusion of those who cannot read and write, and that the researcher cannot couch for the validity of the responses from self-administered surveys. However in our study the response rate was high more than 95 % and also around 95.7% of participants had at least attended primary school level. Four point three (4.3%) percent of the participants did not respond the question on educational background.”

All authors have seen and approved the manuscript, and have contributed to the work. There are no copyright restrictions. No authors have conflicts of interest to disclose.

Thank you for considering our revised manuscript. Please contact me for additional information.

Sincerely,

Lebapotswe B Tlale

Public Health Specialist

Ministry of health and Wellness

Cell: +267 71416607 and +267 73340732

Email: cypro330@yahoo.com

---

## [Editor Report · Decision Letter 3]

19 Jan 2022

Acceptance Rate and Risk Perception towards the COVID-19 Vaccine in Botswana

PONE-D-21-09673R3

Dear Dr. Tlale,

We’re pleased to inform you that your manuscript has been judged scientifically suitable for publication and will be formally accepted for publication once it meets all outstanding technical requirements.

Kind regards,

David Teye Doku

Academic Editor

PLOS ONE
---

## [Editor Report · Acceptance letter]

28 Jan 2022

PONE-D-21-09673R3 

Acceptance Rate and Risk Perception towards the COVID-19 Vaccine in Botswana 

Dear Dr. Tlale:

I'm pleased to inform you that your manuscript has been deemed suitable for publication in PLOS ONE. Congratulations! Your manuscript is now with our production department. 

Kind regards, 

on behalf of

Dr. David Teye Doku 

Academic Editor

PLOS ONE